# Fast Approximation of Similarity Graphs with Kernel Density Estimation

**Peter Macgregor**
School of Informatics
University of Edinburgh
United Kingdom

**He Sun**
School of Informatics
University of Edinburgh
United Kingdom

## Abstract

Constructing a similarity graph from a set $X$ of data points in $\mathbb{R}^d$ is the first step of many modern clustering algorithms. However, typical constructions of a similarity graph have high time complexity, and a quadratic space dependency with respect to $|X|$. We address this limitation and present a new algorithmic framework that constructs a sparse approximation of the fully connected similarity graph while preserving its cluster structure. Our presented algorithm is based on the kernel density estimation problem, and is applicable for arbitrary kernel functions. We compare our designed algorithm with the well-known implementations from the `scikit-learn` library and the `FAISS` library, and find that our method significantly outperforms the implementation from both libraries on a variety of datasets.

## 1 Introduction

Given a set $X = \{x_1, \ldots, x_n\} \subset \mathbb{R}^d$ of data points and a similarity function $k : \mathbb{R}^d \times \mathbb{R}^d \to \mathbb{R}_{\geq 0}$ for any pair of data points $x_i$ and $x_j$, the objective of clustering is to partition these $n$ data points into clusters such that similar points are in the same cluster. As a fundamental data analysis technique, clustering has been extensively studied in different disciplines ranging from algorithms and machine learning, social network analysis, to data science and statistics.

One prominent approach for clustering data points in Euclidean space consists of two simple steps: the first step is to construct a *similarity graph* $\mathsf{K} = (V, E, w)$ from $X$, where every vertex $v_i$ of G corresponds to $x_i \in X$, and different vertices $v_i$ and $v_j$ are connected by an edge with weight $w(v_i, v_j)$ if their similarity $k(x_i, x_j)$ is positive, or higher than some threshold. Secondly, we apply spectral clustering on G, and its output naturally corresponds to some clustering on $X$ [19]. Because of its out-performance over traditional clustering algorithms like $k$-means, this approach has become one of the most popular modern clustering algorithms.

On the other side, different constructions of similarity graphs have significant impact on the quality and time complexity of spectral clustering, which is clearly acknowledged and appropriately discussed by von Luxburg [31]. Generally speaking, there are two types of similarity graphs:

- the first one is the $k$-nearest neighbour graph ($k$-NN graph), in which every vertex $v_i$ connects to $v_j$ if $v_j$ is among the $k$-nearest neighbours of $v_i$. A $k$-NN graph is sparse by construction, but loses some of the structural information in the dataset since $k$ is usually small and the added edges are unweighted.

- the second one is the fully connected graph, in which different vertices $v_i$ and $v_j$ are connected with weight $w(v_i, v_j) = k(x_i, x_j)$. While a fully connected graph maintains most of the distance information about $X$, this graph is dense and storing such graphs requires *quadratic* memory in $n$.

37th Conference on Neural Information Processing Systems (NeurIPS 2023).

Taking the pros and cons of the two constructions into account, one would naturally ask the question:

*Is it possible to directly construct a sparse graph that preserves the cluster structure of a fully connected similarity graph?*

We answer this question affirmatively, and present a fast algorithm that constructs an approximation of the fully connected similarity graph. Our constructed graph consists of only $\widetilde{O}(n)$ edges[1], and preserves the cluster structure of the fully connected similarity graph.

## 1.1 Our Result

Given any set $X = \{x_1, \ldots, x_n\} \subset \mathbb{R}^d$ and a kernel function $k : \mathbb{R}^d \times \mathbb{R}^d \to \mathbb{R}_{\geq 0}$, a fully connected similarity graph $\mathsf{K} = (V, E, w)$ of $X$ consists of $n$ vertices, and every $v_i \in V$ corresponds to $x_i \in X$; we set $w(v_i, v_j) \triangleq k(x_i, x_j)$ for any different $v_i$ and $v_j$. We introduce an efficient algorithm that constructs a sparse graph $\mathsf{G}$ *directly* from $X$, such that $\mathsf{K}$ and $\mathsf{G}$ share the same cluster-structure, and the graph matrices for $\mathsf{K}$ and $\mathsf{G}$ have approximately the same eigen-gap. This ensures that spectral clustering from $\mathsf{G}$ and $\mathsf{K}$ return approximately the same result.

The design of our algorithm is based on a novel reduction from the approximate construction of similarity graphs to the problem of Kernel Density Estimation (KDE). This reduction shows that any algorithm for the KDE can be employed to construct a sparse representation of a fully connected similarity graph, while preserving the cluster-structure of the input data points. This is summarised as follows:

**Theorem 1** (Informal Statement of Theorem 2). *Given a set of data points $X = \{x_1, \ldots, x_n\} \subset \mathbb{R}^d$ as input, there is a randomised algorithm that constructs a sparse graph $\mathsf{G}$ of $X$, such that it holds with probability at least $9/10$ that*

1. *graph $\mathsf{G}$ has $\widetilde{O}(n)$ edges,*

2. *graph $\mathsf{G}$ has the same cluster structure as the fully connected similarity graph $\mathsf{K}$ of $X$.*

*The algorithm uses an approximate KDE algorithm as a black-box, and has running time $\widetilde{O}(T_{\mathsf{KDE}}(n, n, \epsilon))$ for $\epsilon \leq 1/(6 \log(n))$, where $T_{\mathsf{KDE}}(n, n, \epsilon)$ is the running time of solving the KDE problem for $n$ data points up to a $(1 + \epsilon)$-approximation.*

This result builds a novel connection between the KDE and the fast construction of similarity graphs, and further represents a state-of-the-art algorithm for constructing similarity graphs. For instance, when employing the fast Gauss transform [10] as the KDE solver, Theorem 1 shows that a sparse representation of the fully connected similarity graph with the Gaussian kernel can be constructed in $\widetilde{O}(n)$ time when $d$ is constant. As such, in the case of low dimensions, spectral clustering runs as fast (up to a poly-logarithmic factor) as the time needed to read the input data points. Moreover, any improved algorithm for the KDE would result in a faster construction of approximate similarity graphs.

To demonstrate the significance of this work in practice, we compare the performance of our algorithm with five competing algorithms from the well-known `scikit-learn` library [20] and `FAISS` library [11]: the algorithm that constructs a fully connected Gaussian kernel graph, and four algorithms that construct different variants of $k$-nearest neighbour graphs. We apply spectral clustering on the six constructed similarity graphs, and compare the quality of the resulting clustering. For a typical input dataset of 15,000 points in $\mathbb{R}^2$, our algorithm runs in 4.7 seconds, in comparison with between 16.1 – 128.9 seconds for the five competing algorithms from `scikit-learn` and `FAISS` libraries. As shown in Figure 1, all the six algorithms return reasonable output.

We further compare the quality of the six algorithms on the BSDS image segmentation dataset [2], and our algorithm presents clear improvement over the other five algorithms based on the output's average Rand Index. In particular, due to its *quadratic* memory requirement in the input size, one would need to reduce the resolution of every image down to 20,000 pixels in order to construct the fully connected similarity graph with `scikit-learn` on a typical laptop. In contrast, our algorithm is able to segment the full-resolution image with over 150,000 pixels. Our experimental result on

---

[1]We use $\widetilde{O}(n)$ to represent $O\left(n \cdot \log^c(n)\right)$ for some constant $c$.

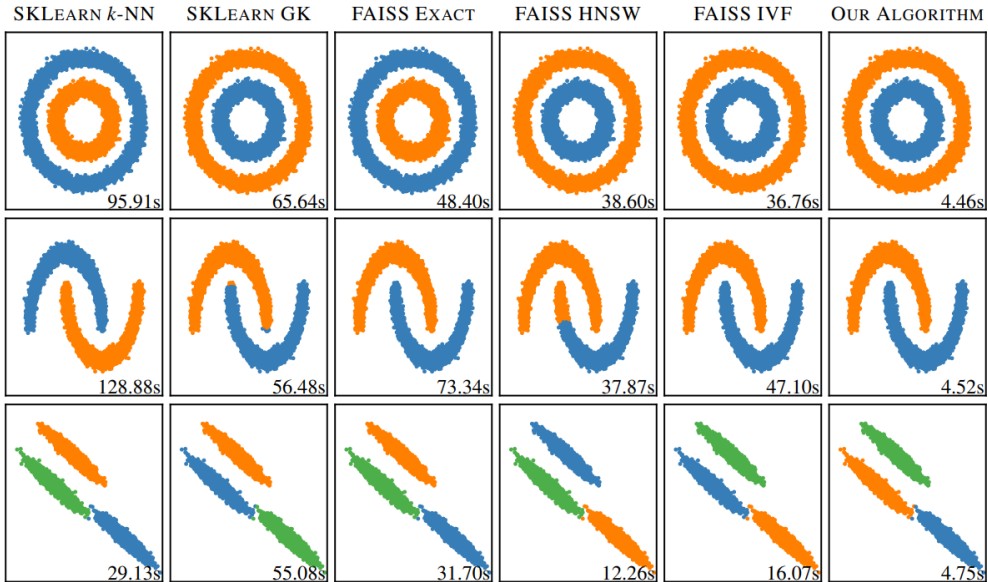

| SKLearn $k$-NN | SKLearn GK | FAISS Exact | FAISS HNSW | FAISS IVF | Our Algorithm |

95.91s  65.64s  48.40s  38.60s  36.76s  4.46s

128.88s  56.48s  73.34s  37.87s  47.10s  4.52s

29.13s  55.08s  31.70s  12.26s  16.07s  4.75s

Figure 1: Output of spectral clustering with different similarity graph constructions.

the BSDS dataset is showcased in Figure 2 and demonstrates that, in comparison with SKLEARN GK, our algorithm identifies a more detailed pattern on the butterfly's wing. In contrast, none of the $k$-nearest neighbour based algorithms from the two libraries is able to identify the wings of the butterfly.

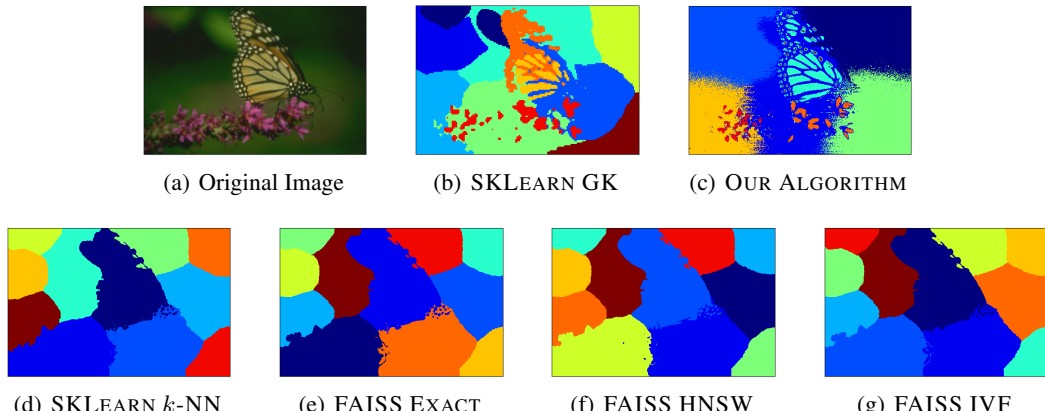

(a) Original Image     (b) SKLEARN GK     (c) OUR ALGORITHM

(d) SKLEARN $k$-NN    (e) FAISS EXACT    (f) FAISS HNSW    (g) FAISS IVF

Figure 2: Comparison on the performance of spectral clustering with different similarity graph constructions. Here, SKLEARN GK is based on the fully connected similarity graph construction, and (d) – (g) are based on different $k$-nearest neighbour graph constructions from the two libraries.

## 1.2 Related Work

There are a number of works on efficient constructions of $\varepsilon$-neighbourhood graphs and $k$-NN graphs. For instance, Dong et al. [9] presents an algorithm for approximate $k$-NN graph construction, and their algorithm is based on local search. Wang et al. [32] presents an LSH-based algorithm for constructing an approximate $k$-NN graph, and employs several sampling and hashing techniques to reduce the computational and parallelisation cost. These two algorithms [9, 32] have shown to work very well in practice, but lack a theoretical guarantee on the performance.

Our work also relates to a large and growing number of KDE algorithms. Charikar and Siminelakis [7] study the KDE problem through LSH, and present a class of unbiased estimators for kernel density in high dimensions for a variety of commonly used kernels. Their work has been improved through the sketching technique [24], and a revised description and analysis of the original algorithm [4]. Charikar et al. [6] presents a data structure for the KDE problem, and their result essentially matches the query time and space complexity for most studied kernels in the literature. In addition, there are studies on designing efficient KDE algorithms based on interpolation of kernel density estimators [29], and coresets [12].

Our work further relates to efficient constructions of spectral sparsifiers for kernel graphs. Quanrud [22] studies smooth kernel functions, and shows that an explicit $(1 + \varepsilon)$-approximate spectral approximation of the geometric graph with $\widetilde{O}(n/\varepsilon^2)$ edges can be computed in $\widetilde{O}(n/\varepsilon^2)$ time. Bakshi et al. [5] proves that, under the strong exponential time hypothesis, constructing an $O(1)$-approximate spectral sparsifier with $O(n^{2-\delta})$ edges for the Gaussian kernel graph requires $\Omega\left(n \cdot 2^{\log(1/\tau)^{0.32}}\right)$ time, where $\delta < 0.01$ is a fixed universal constant and $\tau$ is the minimum entry of the kernel matrix. Compared with their results, we show that, when the similarity graph with the Gaussian kernel presents a well-defined structure of clusters, an approximate construction of this similarity graph can be constructed in nearly-linear time.

## 2 Preliminaries

Let $\mathsf{G} = (V, E, w_\mathsf{G})$ be an undirected graph with weight function $w_\mathsf{G} : E \to \mathbb{R}_{\geq 0}$, and $n \triangleq |V|$. The degree of any vertex $v$ is defined as $\deg_\mathsf{G}(v) \triangleq \sum_{u \sim v} w_\mathsf{G}(u, v)$, where we write $u \sim v$ if $\{u, v\} \in E(\mathsf{G})$. For any $S \subset V$, the volume of $S$ is defined by $\mathrm{vol}_\mathsf{G}(S) \triangleq \sum_{v \in S} \deg_\mathsf{G}(v)$, and the conductance of $S$ is defined by

$$\phi_\mathsf{G}(S) \triangleq \frac{\partial_\mathsf{G}(S)}{\mathrm{vol}_\mathsf{G}(S)},$$

where $\partial_\mathsf{G}(S) \triangleq \sum_{u \in S, v \notin S} w_\mathsf{G}(u, v)$. For any $k \geq 2$, we call subsets of vertices $A_1, \ldots, A_k$ a *k-way partition* if $A_i \neq \emptyset$ for any $1 \leq i \leq k$, $A_i \cap A_j = \emptyset$ for any $i \neq j$, and $\bigcup_{i=1}^k A_i = V$. Moreover, we define the *k-way expansion constant* by

$$\rho_\mathsf{G}(k) \triangleq \min_{\text{partition } A_1, \ldots, A_k} \max_{1 \leq i \leq k} \phi_\mathsf{G}(A_i).$$

Note that a lower value of $\rho_\mathsf{G}(k)$ ensures the existence of $k$ clusters $A_1, \ldots, A_k$ of low conductance, i.e, $\mathsf{G}$ has at least $k$ clusters.

For any undirected graph $\mathsf{G}$, the adjacency matrix $\mathbf{A}_\mathsf{G}$ of $\mathsf{G}$ is defined by $\mathbf{A}_\mathsf{G}(u, v) = w_\mathsf{G}(u, v)$ if $u \sim v$, and $\mathbf{A}_\mathsf{G}(u, v) = 0$ otherwise. We write $\mathbf{D}_\mathsf{G}$ as the diagonal matrix defined by $\mathbf{D}_\mathsf{G}(v, v) = \deg_\mathsf{G}(v)$, and the normalised Laplacian of $\mathsf{G}$ is defined by $\mathbf{N}_\mathsf{G} \triangleq \mathbf{I} - \mathbf{D}_\mathsf{G}^{-1/2} \mathbf{A}_\mathsf{G} \mathbf{D}_\mathsf{G}^{-1/2}$. For any PSD matrix $\mathbf{B} \in \mathbb{R}^{n \times n}$, we write the eigenvalues of $\mathbf{B}$ as $\lambda_1(\mathbf{B}) \leq \ldots \leq \lambda_n(\mathbf{B})$.

It is well-known that, while computing $\rho_\mathsf{G}(k)$ exactly is NP-hard, $\rho_\mathsf{G}(k)$ is closely related to $\lambda_k$ through the higher-order Cheeger inequality [13]: it holds for any $k$ that

$$\lambda_k(\mathbf{N}_\mathsf{G})/2 \leq \rho_\mathsf{G}(k) \leq O(k^3)\sqrt{\lambda_k(\mathbf{N}_\mathsf{G})}.$$

### 2.1 Fully Connected Similarity Graphs

We use $X \triangleq \{x_1, \ldots x_n\}$ to represent the set of input data points, where every $x_i \in \mathbb{R}^d$. Given $X$ and some kernel function $k : \mathbb{R}^d \times \mathbb{R}^d \to \mathbb{R}_{\geq 0}$, we use $\mathsf{K} = (V_\mathsf{K}, E_\mathsf{K}, w_\mathsf{K})$ to represent the fully connected similarity graph from $X$, which is constructed as follows: every $v_i \in V_\mathsf{K}$ corresponds to $x_i \in X$, and any pair of different $v_i$ and $v_j$ is connected by an edge with weight $w_\mathsf{K}(v_i, v_j) = k(x_i, x_j)$. One of the most common kernels used for clustering is the Gaussian kernel, which is defined by

$$k(x_i, x_j) = \exp\left(-\frac{\|x_i - x_j\|_2^2}{\sigma^2}\right)$$

for some bandwidth parameter $\sigma$. Other popular kernels include the Laplacian kernel and the exponential kernel which use $\|x_i - x_j\|_1$ and $\|x_i - x_j\|_2$ in the exponent respectively.

## 2.2 Kernel Density Estimation

Our work is based on algorithms for kernel density estimation (KDE), which is defined as follows. Given a kernel function $k : \mathbb{R}^d \times \mathbb{R}^d \to \mathbb{R}_{\geq 0}$ with $n$ source points $x_1, \ldots, x_n \in \mathbb{R}^d$ and $m$ target points $y_1, \ldots, y_m \in \mathbb{R}^d$, the KDE problem is to compute $g_{[1,n]}(y_1), \ldots g_{[1,n]}(y_m)$, where

$$g_{[a,b]}(y_i) \triangleq \sum_{j=a}^{b} k(y_i, x_j) \tag{1}$$

for $1 \leq i \leq m$. While a direct computation of the $m$ values $g_{[1,n]}(y_1), \ldots g_{[1,n]}(y_m)$ requires $mn$ operations, there is substantial research to develop faster algorithms approximating these $m$ quantities.

In this paper we are interested in the algorithms that approximately compute $g_{[1,n]}(y_i)$ for all $1 \leq i \leq m$ up to a $(1 \pm \epsilon)$-multiplicative error, and use $T_{\mathsf{KDE}}(m, n, \epsilon)$ to denote the asymptotic complexity of such a KDE algorithm. We also require that $T_{\mathsf{KDE}}(m, n, \epsilon)$ is superadditive in $m$ and $n$; that is, for $m = m_1 + m_2$ and $n = n_1 + n_2$, we have

$$T_{\mathsf{KDE}}(m_1, n_1, \epsilon) + T_{\mathsf{KDE}}(m_2, n_2, \epsilon) \leq T_{\mathsf{KDE}}(m, n, \epsilon);$$

it is known that such property holds for many KDE algorithms (e.g., [1, 6, 10]).

# 3 Cluster-Preserving Sparsifiers

A graph sparsifier is a sparse representation of an input graph that inherits certain properties of the original dense graph. The efficient construction of sparsifiers plays an important role in designing a number of nearly-linear time graph algorithms. However, most algorithms for constructing sparsifiers rely on the recursive decomposition of an input graph [26], sampling with respect to effective resistances [15, 25], or fast SDP solvers [14]; all of these need the explicit representation of an input graph, requiring $\Omega(n^2)$ time and space complexity for a fully connected graph.

Sun and Zanetti [27] study a variant of graph sparsifiers that mainly preserve the cluster structure of an input graph, and introduce the notion of *cluster-preserving sparsifier* defined as follows:

**Definition 1** (Cluster-preserving sparsifier). Let $\mathsf{K} = (V, E, w_\mathsf{K})$ be any graph, and $\{A_i\}_{i=1}^k$ the $k$-way partition of K corresponding to $\rho_\mathsf{K}(k)$. We call a re-weighted subgraph $\mathsf{G} = (V, F \subset E, w_\mathsf{G})$ a cluster-preserving sparsifier of K if $\phi_\mathsf{G}(A_i) = O(k \cdot \phi_\mathsf{K}(A_i))$ for $1 \leq i \leq k$, and $\lambda_{k+1}(\mathbf{N}_\mathsf{G}) = \Omega(\lambda_{k+1}(\mathbf{N}_\mathsf{K}))$.

Notice that graph K has exactly $k$ clusters if (i) K has $k$ disjoint subsets $A_1, \ldots, A_k$ of low conductance, and (ii) any $(k + 1)$-way partition of K would include some $A \subset V$ of high conductance, which would be implied by a lower bound on $\lambda_{k+1}(\mathbf{N}_\mathsf{K})$ due to the higher-order Cheeger inequality. Together with the well-known eigen-gap heuristic [13, 31] and theoretical analysis on spectral clustering [16, 21], the two properties in Definition 1 ensures that spectral clustering returns approximately the same output from K and H.[2]

Now we present the algorithm in [27] for constructing a cluster-preserving sparsifier, and we call it the SZ algorithm for simplicity. Given any input graph $\mathsf{K} = (V, E, w_\mathsf{K})$ with weight function $w_\mathsf{K}$, the algorithm computes

$$p_u(v) \triangleq \min\left\{ C \cdot \frac{\log n}{\lambda_{k+1}} \cdot \frac{w_\mathsf{K}(u, v)}{\deg_\mathsf{K}(u)}, 1 \right\}, \quad \text{and} \quad p_v(u) \triangleq \min\left\{ C \cdot \frac{\log n}{\lambda_{k+1}} \cdot \frac{w_\mathsf{K}(v, u)}{\deg_\mathsf{K}(v)}, 1 \right\},$$

for every edge $e = \{u, v\}$, where $C \in \mathbb{R}^+$ is some constant. Then, the algorithm samples every edge $e = \{u, v\}$ with probability

$$p_e \triangleq p_u(v) + p_v(u) - p_u(v) \cdot p_v(u),$$

and sets the weight of every sampled $e = \{u, v\}$ in G as $w_\mathsf{G}(u, v) \triangleq w_\mathsf{K}(u, v)/p_e$. By setting $F$ as the set of the sampled edges, the algorithm returns $\mathsf{G} = (V, F, w_\mathsf{G})$ as output. It is shown in [27] that, with high probability, the constructed G has $\widetilde{O}(n)$ edges and is a cluster-preserving sparsifier of K.

---

[2]The most interesting regime for this definition is $k = \widetilde{O}(1)$ and $\lambda_{k+1}(\mathbf{N}_\mathsf{K}) = \Omega(1)$, and we assume this in the rest of the paper.

We remark that a cluster-preserving sparsifier is a much weaker notion than a spectral sparsifier, which approximately preserves all the cut values and the eigenvalues of the graph Laplacian matrices. On the other side, while a cluster-preserving sparsifier is sufficient for the task of graph clustering, the SZ algorithm runs in $\Omega(n^2)$ time for a fully connected input graph, since it's based on the computation of the vertex degrees as well as the sampling probabilities $p_u(v)$ for every pair of vertices $u$ and $v$.

## 4  Algorithm

This section presents our algorithm that directly constructs an approximation of a fully connected similarity graph from $X \subseteq \mathbb{R}^d$ with $|X| = n$. As the main theoretical contribution, we demonstrate that neither the quadratic space complexity for directly constructing a fully connected similarity graph nor the quadratic time complexity of the SZ algorithm is necessary when approximately constructing a fully connected similarity graph for the purpose of clustering. The performance of our algorithm is as follows:

**Theorem 2** (Main Result). *Given a set of data points $X = \{x_1, \ldots, x_n\} \subset \mathbb{R}^d$ as input, there is a randomised algorithm that constructs a sparse graph G of $X$, such that it holds with probability at least $9/10$ that*

1. *graph G has $\widetilde{O}(n)$ edges,*

2. *graph G has the same cluster structure as the fully connected similarity graph K of $X$; that is, if K has $k$ well-defined clusters, then it holds that $\rho_G(k) = O(k \cdot \rho_K(k))$ and $\lambda_{k+1}(\mathbf{N_G}) = \Omega(\lambda_{k+1}(\mathbf{N_K}))$.*

*The algorithm uses an approximate KDE algorithm as a black-box, and has running time $\widetilde{O}(T_{\mathsf{KDE}}(n, n, \epsilon))$ for $\epsilon \leq 1/(6\log(n))$.*

### 4.1  Algorithm Description

At a very high level, our designed algorithm applies a KDE algorithm as a black-box, and constructs a cluster-preserving sparsifier by simultaneous sampling of the edges from a *non-explicitly constructed* fully connected graph. To explain our technique, we first claim that, for an arbitrary $x_i$, a random $x_j$ can be sampled with probability $k(x_i, x_j)/\deg_K(v_i)$ through $O(\log n)$ queries to a KDE algorithm. To see this, notice that we can apply a KDE algorithm to compute the probability that the sampled neighbour is in some set $X_1 \subset X$, i.e.,

$$\mathbb{P}\left[z \in X_1\right] = \sum_{x_j \in X_1} \frac{k(x_i, x_j)}{\deg_K(v_i)} = \frac{g_{X_1}(x_i)}{g_X(x_i)},$$

where we use $g_X(y_i)$ to denote that the KDE is taken with respect to the set of source points $X$. Based on this, we recursively split the set of possible neighbours in half and choose between the two subsets with the correct probability. The sampling procedure is summarised as follows, and is illustrated in Figure 3. We remark that the method of sampling a random neighbour of a vertex in K through KDE and binary search also appears in Backurs et al. [5]

1. Set the feasible neighbours to be $X = \{x_1, \ldots, x_n\}$.
2. While $|X| > 1$:
    - Split $X$ into $X_1$ and $X_2$ with $|X_1| = \lfloor|X|/2\rfloor$ and $|X_2| = \lceil|X|/2\rceil$.
    - Compute $g_X(x_i)$ and $g_{X_1}(x_i)$; set $X \leftarrow X_1$ with probability $g_{X_1}(x_i)/g_X(x_i)$, and $X \leftarrow X_2$ with probability $1 - g_{X_1}(x_i)/g_X(x_i)$.
3. Return the remaining element in $X$ as the sampled neighbour.

Next we generalise this idea and show that, instead of sampling a neighbour of one vertex at a time, a KDE algorithm allows us to sample neighbours of every vertex in the graph "simultaneously". Our designed sampling procedure is formalised in Algorithm 1.

Finally, to construct a cluster-preserving sparsifier, we apply Algorithm 1 to sample $O(\log n)$ neighbours for every vertex $v_i$, and set the weight of every sampled edge $v_i \sim v_j$ as

$$w_G(v_i, v_j) = \frac{k(x_i, x_j)}{\widehat{p}(i, j)}, \tag{2}$$

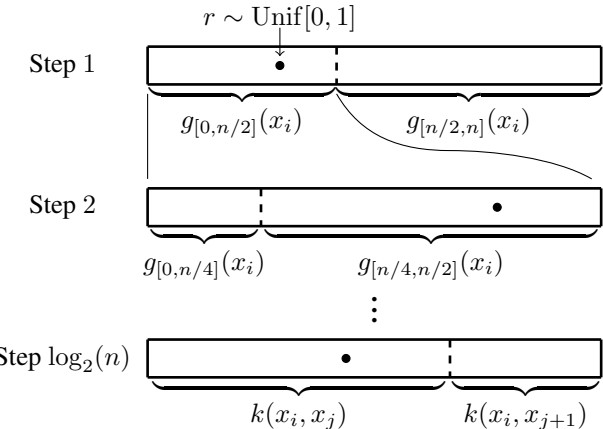

Figure 3: The procedure of sampling a neighbour $v_j$ of $v_i$ with probability $k(x_i, x_j)/\deg_K(v_i)$. Our algorithm performs a binary search to find the sampled neighbour. At each step, the value of two kernel density estimates are used to determine where the sample lies. Notice that the algorithm doesn't compute any edge weights directly until the last step.

---

**Algorithm 1** SAMPLE

1: **Input:** set $S$ of $\{y_i\}$
        set $X$ of $\{x_i\}$
2: **Output:**
        $E = \{(y_i, x_j)$ for some $i$ and $j\}$
3: **if** $|X| = 1$ **then**
4:     **return** $S \times X$
5: **else**
6:     $X_1 = \{x_j : j < |X|/2\}$
7:     $X_2 = \{x_j : j \geq |X|/2\}$
8:     Compute $g_{X_1}(y_i)$ for all $i$ with a KDE algorithm
9:     Compute $g_{X_2}(y_i)$ for all $i$ with a KDE algorithm
10:     $S_1 = S_2 = \emptyset$
11:     **for** $y_i \in S$ **do**
12:         $r \sim \mathrm{Unif}[0, 1]$
13:         **if** $r \leq g_{X_1}(y_i)/(g_{X_1}(y_i) + g_{X_2}(y_i))$ **then**
14:             $S_1 = S_1 \cup \{y_i\}$
15:         **else**
16:             $S_2 = S_2 \cup \{y_i\}$
17:         **end if**
18:     **end for**
19:     **return** SAMPLE$(S_1, X_1) \cup$ SAMPLE$(S_2, X_2)$
20: **end if**

---

where $\widehat{p}(i, j) \triangleq \widehat{p}_i(j) + \widehat{p}_j(i) - \widehat{p}_i(j) \cdot \widehat{p}_j(i)$ is an estimate of the sampling probability of edge $v_i \sim v_j$, and

$$\widehat{p}_i(j) \triangleq \min\left\{6C \cdot \log n \cdot \frac{k(x_i, x_j)}{g_{[1,n]}(x_i)}, 1\right\}$$

for some constant $C \in \mathbb{R}^+$; see Algorithm 2 for the formal description.

### 4.2 Algorithm Analysis

Now we analyse Algorithm 2, and sketch the proof of Theorem 2. We first analyse the running time of Algorithm 2. Since it involves $O(\log n)$ executions of Algorithm 1 in total, it is sufficient to examine the running time of Algorithm 1.

We visualise the recursion of Algorithm 1 with respect to $S$ and $X$ in Figure 4. Notice that, although the number of nodes doubles at each level of the recursion tree, the total number of samples $S$ and data points $X$ remain constant for each level of the tree: it holds for any $i$ that $\bigcup_{j=1}^{2^i} S_{i,j} = S_0$ and

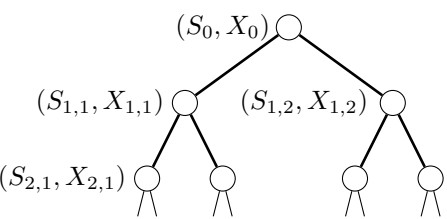

Figure 4: The recursion tree for Algorithm 1.

---

**Algorithm 2** FASTSIMILARITYGRAPH

---

1: **Input:** data point set $X = \{x_1, \ldots, x_n\}$
2: **Output:** similarity graph $\mathsf{G}$
3: $E = \emptyset$, $L = 6C \cdot \log(n)/\lambda_{k+1}$
4: **for** $\ell \in [1, L]$ **do**
5:    $E = E \cup \textsc{Sample}(X, X)$
6: **end for**
7: Compute $g_{[1,n]}(x_i)$ for each $i$ with a KDE algorithm

8: **for** $(v_i, v_j) \in E$ **do**
9:    $\widehat{p}_i(j) = \min\left\{L \cdot k(x_i, x_j)/g_{[1,n]}(x_i), 1\right\}$
10:   $\widehat{p}_j(i) = \min\left\{L \cdot k(x_i, x_j)/g_{[1,n]}(x_j), 1\right\}$
11:   $\widehat{p}(i, j) = \widehat{p}_i(j) + \widehat{p}_j(i) - \widehat{p}_i(j) \cdot \widehat{p}_j(i)$
12:   Set $w_{\mathsf{G}}(v_i, v_j) = k(x_i, x_j)/\widehat{p}(i, j)$
13: **end for**
14: **return** graph $\mathsf{G} = (X, E, w_{\mathsf{G}})$

---

$\bigcup_{j=1}^{2^i} X_{i,j} = X_0$. Since the running time of the KDE algorithm is superadditive, the combined running time of all nodes at level $i$ of the tree is

$$T_i = \sum_{j=1}^{2^i} T_{\mathsf{KDE}}(|S_{i,j}|, |X_{i,j}|, \epsilon) \leq T_{\mathsf{KDE}}\left(\sum_{j=1}^{2^i} |S_{i,j}|, \sum_{j=1}^{2^i} |X_{i,j}|, \epsilon\right) = T_{\mathsf{KDE}}(n, n, \epsilon).$$

Hence, the total running time of Algorithm 2 is $\widetilde{O}(T_{\mathsf{KDE}}(n, n, \epsilon))$ as the tree has $\log_2(n)$ levels. Moreover, when applying the *Fast Gauss Transform* (FGT) [10] as the KDE algorithm, the total running time of Algorithm 1 is $\widetilde{O}(n)$ when $d = O(1)$.

Finally, we prove that the output of Algorithm 2 is a cluster preserving sparsifier of a fully connected similarity graph, and has $\widetilde{O}(n)$ edges. Notice that, comparing with the sampling probabilities $p_u(v)$ and $p_v(u)$ used in the SZ algorithm, Algorithm 2 samples each edge through $O(\log n)$ recursive executions of a KDE algorithm, each of which introduces a $(1 + \epsilon)$-multiplicative error. We carefully examine these $(1 + \epsilon)$-multiplicative errors and prove that the actual sampling probability $\widetilde{p}(i, j)$ used in Algorithm 2 is an $O(1)$-approximation of $p_e$ for every $e = \{v_i, v_j\}$. As such the output of Algorithm 2 is a cluster-preserving sparsifier of a fully connected similarity graph, and satisfies the two properties of Theorem 2. The total number of edges in $\mathsf{G}$ follows from the sampling scheme. We refer the reader to Appendix A for the complete proof of Theorem 2.

## 5 Experiments

In this section, we empirically evaluate the performance of spectral clustering with our new algorithm for constructing similarity graphs. We compare our algorithm with the algorithms for similarity graph construction provided by the `scikit-learn` library [20] and the FAISS library [11] which are commonly used machine learning libraries for Python. In the remainder of this section, we compare the following six spectral clustering methods.

1. SKLEARN GK: spectral clustering with the fully connected Gaussian kernel similarity graph constructed with the `scikit-learn` library.
2. SKLEARN $k$-NN: spectral clustering with the $k$-nearest neighbour similarity graph constructed with the `scikit-learn` library.
3. FAISS EXACT: spectral clustering with the exact $k$-nearest neighbour graph constructed with the FAISS library.
4. FAISS HNSW: spectral clustering with the approximate $k$-nearest neighbour graph constructed with the "Hierarchical Navigable Small World" method [18].
5. FAISS IVF: spectral clustering with the approximate $k$-nearest neighbour graph constructed with the "Invertex File Index" method [3].
6. OUR ALGORITHM: spectral clustering with the Gaussian kernel similarity graph constructed by Algorithm 2.

We implement Algorithm 2 in C++, using the implementation of the Fast Gauss Transform provided by Yang et al. [33], and use the Python SciPy [30] and `stag` [17] libraries for eigenvector computation and graph operations respectively. The `scikit-learn` and FAISS libraries are well-optimised

and use C, C++, and FORTRAN for efficient implementation of core algorithms. We first employ classical synthetic clustering datasets to clearly compare how the running time of different algorithms is affected by the number of data points. This experiment highlights the nearly-linear time complexity of our algorithm. Next we demonstrate the applicability of our new algorithm for image segmentation on the Berkeley Image Segmentation Dataset (BSDS) [2].

For each experiment, we set $k = 10$ for the approximate nearest neighbour algorithms. For the synthetic datasets, we set the $\sigma$ value of the Gaussian kernel used by SKLEARN GK and OUR ALGORITHM to be $0.1$, and for the BSDS experiment we set $\sigma = 0.2$. This choice follows the heuristic suggested by von Luxburg [31] to choose $\sigma$ to approximate the distance from a point to its $k$th nearest neighbour. All experiments are performed on an HP ZBook laptop with an 11th Gen Intel(R) Core(TM) i7-11800H @ 2.30GHz processor and 32 GB RAM. The code to reproduce our results is available at `https://github.com/pmacg/kde-similarity-graph`.

## 5.1 Synthetic Dataset

In this experiment we use the `scikit-learn` library to generate 15,000 data points in $\mathbb{R}^2$ from a variety of classical synthetic clustering datasets. The data is generated with the `make_circles`, `make_moons`, and `make_blobs` methods of the `scikit-learn` library with a noise parameter of $0.05$. The linear transformation $\{\{0.6, -0.6\}, \{-0.4, 0.8\}\}$ is applied to the blobs data to create asymmetric clusters. As shown in Figure 1, all of the algorithms return approximately the same clustering, but our algorithm runs much faster than the ones from `scikit-learn` and `FAISS`.

We further compare the speedup of our algorithm against the five competitors on the `two_moons` dataset with an increasing number of data points, and our result is reported in Figure 5. Notice that the running time of the `scikit-learn` and `FAISS` algorithms scales roughly quadratically with the size of the input, while the running time of our algorithm is nearly linear. Furthermore, we note that on a laptop with 32 GB RAM, the SKLEARN GK algorithm could not scale beyond 20,000 data points due to its quadratic memory requirement, while our new algorithm can cluster 1,000,000 data points in 240 seconds under the same memory constraint.

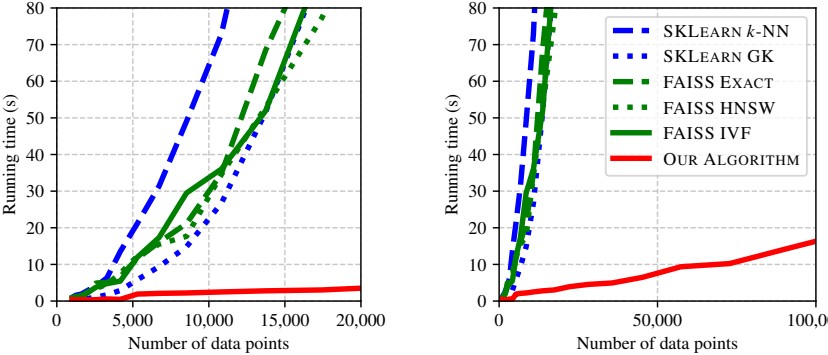

Figure 5: Comparison of the running time of spectral clustering on the two moons dataset. In every case, all algorithms return the correct clusters.

## 5.2 BSDS Image Segmentation Dataset

Finally we study the application of spectral clustering for image segmentation on the BSDS dataset. The dataset contains $500$ images with several ground-truth segmentations for each image. Given an input image, we consider each pixel to be a point $(r, g, b, x, y)^\mathsf{T} \in \mathbb{R}^5$ where $r$, $g$, $b$ are the red, green, blue pixel values and $x$, $y$ are the coordinates of the pixel in the image. Then, we construct a similarity graph based on these points, and apply spectral clustering, for which we set the number of clusters to be the median number of clusters in the provided ground truth segmentations. Our experimental result is reported in Table 1, where the "Downsampled" version is employed to reduce the resolution of the image to 20,000 pixels in order to run the SKLEARN GK and the "Full Resolution" one is to apply the original image of over 150,000 pixels as input. This set of experiments demonstrates our algorithm produces better clustering results with repsect to the average Rand Index [23].

Table 1: The average Rand Index of the output from the six algorithms. Due to the quadratic space complexity constraint, the SKLEARN GK cannot be applied to the full resolution image.

| Algorithm | Downsampled | Full Resolution |
|-----------|-------------|-----------------|
| SKLEARN GK | 0.681 | - |
| SKLEARN $k$-NN | 0.675 | 0.682 |
| FAISS EXACT | 0.675 | 0.680 |
| FAISS HNSW | 0.679 | 0.677 |
| FAISS IVF | 0.675 | 0.680 |
| OUR ALGORITHM | 0.680 | 0.702 |

## 6  Conclusion

In this paper we develop a new technique that constructs a similarity graph, and show that an approximation algorithm for the KDE can be employed to construct a similarity graph with proven approximation guarantee. Applying the publicly available implementations of the KDE as a black-box, our algorithm outperforms five competing ones from the well-known `scikit-learn` and `FAISS` libraries for the classical datasets of low dimensions. We believe that our work will motivate more research on faster algorithms for the KDE in higher dimensions and their efficient implementation, as well as more efficient constructions of similarity graphs.

## Acknowledgements

We would like to thank the anonymous reviewers for their valuable comments on the paper. This work is supported by an EPSRC Early Career Fellowship (EP/T00729X/1).

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

# A   Proof of Theorem 2

This section presents the complete proof of Theorem 2. Let $y_{i,1}, \ldots, y_{i,L}$ be random variables which are equal to the indices of the $L$ points sampled for $x_i$. Recall that by the $\mathsf{SZ}$ algorithm, the "ideal" sampling probability for $x_j$ from $x_i$ is

$$p_i(j) \triangleq \min \left\{ C \cdot \frac{\log(n)}{\lambda_{k+1}} \cdot \frac{k(x_i, x_j)}{\deg_{\mathsf{K}}(v_i)}, 1 \right\}.$$

We denote the actual sampling probability that $x_j$ is sampled from $x_i$ under Algorithm 2 to be

$$\widetilde{p}_i(j) \triangleq \mathbb{P}\left[x_j \in \{y_{i,1}, \ldots y_{i,L}\}\right].$$

Finally, for each added edge, Algorithm 2 also computes an estimate of $p_i(x_j)$ which we denote

$$\widehat{p}_i(j) \triangleq \min \left\{ 6C \cdot \frac{\log(n)}{\lambda_{k+1}} \cdot \frac{k(x_i, x_j)}{g_{[1,n]}(x_i)}, 1 \right\}.$$

Similar with the definition of $p_e$ in Section 3, we define

- $p(i, j) = p_i(j) + p_j(i) - p_i(j) \cdot p_j(i)$,
- $\widetilde{p}(i, j) = \widetilde{p}_i(j) + \widetilde{p}_j(i) - \widetilde{p}_i(j) \cdot \widetilde{p}_j(i)$, and
- $\widehat{p}(i, j) = \widehat{p}_i(j) + \widehat{p}_j(i) - \widehat{p}_i(j) \cdot \widehat{p}_j(i)$.

Notice that, following the convention of [27], we use $p_i(j)$ to refer to the probability that a given edge is sampled *from the vertex* $x_i$ and $p(i, j)$ is the probability that the given edge $\{v_i, v_j\}$ is sampled at all by the algorithm. We use the same convention for $\widetilde{p}_i(j)$ and $\widehat{p}_i(j)$.

We first prove a sequence of lemmas showing that these probabilities are all within a constant factor of each other.

**Lemma 1.** *For any point $x_i$, the probability that a given sampled neighbour $y_{i,l}$ is equal to $j$ is given by*

$$\frac{k(x_i, x_j)}{2 \deg_{\mathsf{K}}(v_i)} \leq \mathbb{P}\left[y_{i,l} = j\right] \leq \frac{2k(x_i, x_j)}{\deg_{\mathsf{K}}(v_i)}.$$

*Proof.* Let $X = \{x_1, \ldots, x_n\}$ be the input data points for Algorithm 2, and $[n] = \{1, \ldots n\}$ be the indices of the input data points. Furthermore, let $[a, b] = \{a, \ldots, b\}$ be the set of indices between $a$ and $b$. Then, in each recursive call to Algorithm 1, we are given a range of indices $[a, b]$ as input and assign $y_{i,l}$ to one half of it: either $[a, \lfloor b/2 \rfloor]$ or $[\lfloor b/a \rfloor + 1, b]$. By Algorithm 1, we have that the probability of assigning $y_{i,l}$ to $[a, \lfloor b/2 \rfloor]$ is

$$\mathbb{P}\left[y_{i,l} \in [a, \lfloor b/2 \rfloor] \mid y_{i,l} \in [a, b]\right] = \frac{g_{[a, \lfloor b/2 \rfloor]}(x_i)}{g_{[a,b]}(x_i)}.$$

By the performance guarantee of the KDE algorithm, we have that $g_{[a,b]}(x_i) \in (1 \pm \epsilon) \deg_{[a,b]}(v_i)$, where we define

$$\deg_{[a,b]}(x_i) \triangleq \sum_{j=a}^{b} k(x_i, x_j).$$

This gives

$$\left( \frac{1 - \epsilon}{1 + \epsilon} \right) \frac{\deg_{[a, \lfloor b/2 \rfloor]}(v_i)}{\deg_{[a,b]}(v_i)} \leq \mathbb{P}\left[y_{i,l} \in X_{[a, \lfloor b/2 \rfloor]} \mid y_{i,l} \in X_{[a,b]}\right] \leq \left( \frac{1 + \epsilon}{1 - \epsilon} \right) \frac{\deg_{[a, \lfloor b/2 \rfloor]}(v_i)}{\deg_{[a,b]}(v_i)}. \quad (3)$$

Next, notice that we can write for a sequence of intervals $[a_1, b_1] \subset [a_2, b_2] \subset \ldots \subset [1, n]$ that

$$\mathbb{P}\left[y_{i,l} = j\right] = \mathbb{P}\left[y_{i,l} = j | y_{i,l} \in [a_1, b_1]\right] \times \mathbb{P}\left[y_{i,l} \in [a_1, b_1] | y_{i,l} \in [a_2, b_2]\right]$$
$$\times \ldots \times \mathbb{P}\left[y_{i,l} \in [a_k, b_k] | y_{i,l} \in [1, n]\right],$$

where each term corresponds to one level of recursion of Algorithm 1 and there are at most $\lceil \log_2(n) \rceil$ terms. Then, by (3) and the fact that the denominator and numerator of adjacent terms cancel out, we have

$$\left( \frac{1-\epsilon}{1+\epsilon} \right)^{\lceil \log_2(n) \rceil} \frac{k(x_i, x_j)}{\deg_{\mathsf{K}}(v_i)} \leq \mathbb{P}\left[ y_{i,l} = j \right] \leq \left( \frac{1+\epsilon}{1-\epsilon} \right)^{\lceil \log_2(n) \rceil} \frac{k(x_i, x_j)}{\deg_{\mathsf{K}}(v_i)}$$

since $\deg_{[j,j]}(v_i) = k(x_i, x_j)$ and $\deg_{[1,n]}(v_i) = \deg_{\mathsf{K}}(v_i)$.

For the lower bound, we have that

$$\left( \frac{1-\epsilon}{1+\epsilon} \right)^{\lceil \log_2(n) \rceil} \geq (1 - 2\epsilon)^{\lceil \log_2(n) \rceil} \geq 1 - 3\log_2(n)\epsilon \geq 1/2,$$

where the final inequality follows by the condition of $\epsilon$ that $\epsilon \leq 1/(6\log_2(n))$.

For the upper bound, we similarly have

$$\left( \frac{1+\epsilon}{1-\epsilon} \right)^{\lceil \log_2(n) \rceil} \leq (1 + 3\epsilon)^{\lceil \log_2(n) \rceil} \leq \exp\left( 3\lceil \log_2(n) \rceil \epsilon \right) \leq e^{2/3} \leq 2,$$

where the first inequality follows since $\epsilon < 1/(6\log_2(n))$. $\qquad\square$

The next lemma shows that the probability of sampling each edge $\{v_i, v_j\}$ is approximately equal to the "ideal" sampling probability $p_i(j)$.

**Lemma 2.** *For every $i$ and $j \neq i$, we have*

$$\frac{9}{10} \cdot p_i(j) \leq \widetilde{p}_i(j) \leq 12 \cdot p_i(j).$$

*Proof.* Let $Y = \{y_{i,1}, \ldots, y_{i,L}\}$ be the neighbours of $x_i$ sampled by Algorithm 2, where $L = 6C \log(n)/\lambda_{k+1}$. Then,

$$\mathbb{P}\left[ j \in Y \right] = 1 - \prod_{l=1}^{L} (1 - \mathbb{P}\left[ y_{i,l} = j \right]) \geq 1 - \left( 1 - \frac{k(x_i, x_j)}{2\deg_{\mathsf{K}}(v_i)} \right)^L \geq 1 - \exp\left( -L \cdot \frac{k(x_i, x_j)}{2\deg_{\mathsf{K}}(v_i)} \right)$$

The proof proceeds by case distinction.

**Case 1:** $p_i(j) \leq 9/10$. In this case, we have

$$\mathbb{P}\left[ j \in Y \right] \geq 1 - \exp\left( -6p_i(j)/2 \right) \geq p_i(j).$$

**Case 2:** $p_i(j) > 9/10$. In this case, we have

$$\mathbb{P}\left[ j \in Y \right] \geq 1 - \exp\left( -\frac{9 \cdot 6}{20} \right) \geq \frac{9}{10},$$

which completes the proof on the lower bound of $\widetilde{p}_i(j)$.

For the upper bound, we have

$$\mathbb{P}\left[ j \in Y \right] \leq 1 - \left( 1 - \frac{2k(x_i, x_j)}{\deg_{\mathsf{K}}(v_i)} \right)^L \leq \frac{2k(x_i, x_j)}{\deg_{\mathsf{K}}(v_i)} \cdot L = 12C \cdot \frac{\log(n)}{\lambda_{k+1}} \cdot \frac{k(x_i, x_j)}{\deg_{\mathsf{K}}(v_i)},$$

from which the statement follows. $\qquad\square$

An immediate corollary of Lemma 2 is as follows.

**Corollary 1.** *For all different $i, j \in [n]$, it holds that*

$$\frac{9}{10} \cdot p(i, j) \leq \widetilde{p}(i, j) \leq 12 \cdot p(i, j)$$

*and*

$$\frac{6}{7} \cdot p(i, j) \leq \widehat{p}(i, j) \leq \frac{36}{5} \cdot p(i, j).$$

*Proof.* For the upper bound of the first statement, we can assume that $p_i(j) \leq 1/12$ and $p_j(i) \leq 1/12$, since otherwise we have $\widetilde{p}(i,j) \leq 1 \leq 12 \cdot p(i,j)$ and the statement holds trivially. Then, by Lemma 2, we have

$$
\begin{aligned}
\widetilde{p}(i,j) &= \widetilde{p}_i(j) + \widetilde{p}_j(i) - \widetilde{p}_i(j) \cdot \widetilde{p}_j(i) \\
&\leq 12 p_i(j) + 12 p_j(i) - 12 p_i(j) \cdot 12 p_j(i) \\
&\leq 12 \left( p_i(j) + p_j(i) - p_i(j) \cdot p_j(i) \right) \\
&= 12 \cdot p(i,j)
\end{aligned}
$$

and

$$
\begin{aligned}
\widetilde{p}(i,j) &= \widetilde{p}_i(j) + \widetilde{p}_j(i) - \widetilde{p}_i(j) \cdot \widetilde{p}_j(i) \\
&\geq \frac{9}{10} \cdot p_i(j) + \frac{9}{10} \cdot p_j(i) - \frac{9}{10} p_i(j) \cdot \frac{9}{10} p_j(i) \\
&\geq \frac{9}{10} \left( p_i(j) + p_j(i) - p_i(j) p_j(i) \right) \\
&= \frac{9}{10} \cdot p(i,j).
\end{aligned}
$$

For the second statement, notice that

$$
\begin{aligned}
\widehat{p}_i(j) &= \min \left\{ \frac{6C \log(n)}{\lambda_{k+1}} \cdot \frac{k(i,j)}{g_{[1,n]}(x_i)}, 1 \right\} \\
&\geq \min \left\{ \frac{1}{1+\varepsilon} \frac{C \log(n)}{\lambda_{k+1}} \cdot \frac{k(i,j)}{\deg_{\mathsf{K}}(v_i)}, 1 \right\} \\
&\geq \frac{1}{1+\varepsilon} \cdot \min \left\{ \frac{C \log(n)}{\lambda_{k+1}} \cdot \frac{k(i,j)}{\deg_{\mathsf{K}}(v_i)}, 1 \right\} \\
&= \frac{1}{1+\varepsilon} \cdot p_i(j) \\
&\geq \frac{6}{7} \cdot p_i(j),
\end{aligned}
$$

since $g_{[1,n]}(x_i)$ is a $(1 \pm \varepsilon)$ approximation of $\deg_{\mathsf{K}}(v_i)$ and $\varepsilon \leq 1/6$. Similarly,

$$
\widehat{p}_i(j) \leq \frac{6}{1-\varepsilon} \cdot p_i(j) \leq \frac{36}{5} \cdot p_i(j).
$$

Then, for the upper bound of the second statement, we can assume that $p_i(j) \leq 5/36$ and $p_j(i) \leq 5/36$, since otherwise $\widehat{p}(i,j) \leq 1 \leq (36/5) \cdot \widetilde{p}(i,j)$ and the statement holds trivially. This implies that

$$
\begin{aligned}
\widehat{p}(i,j) &= \widehat{p}_i(j) + \widehat{p}_j(i) - \widehat{p}_i(j) \cdot \widehat{p}_j(i) \\
&\leq \frac{36}{5} p_i(j) + \frac{36}{5} p_j(i) - \frac{36}{5} p_i(j) \cdot \frac{36}{5} p_j(i) \\
&\leq \frac{36}{5} \left( p_i(j) + p_j(i) - p_i(j) \cdot p_j(i) \right) \\
&= \frac{36}{5} \cdot p(i,j)
\end{aligned}
$$

and

$$
\begin{aligned}
\widehat{p}(i,j) &\geq \frac{6}{7} p_i(j) + \frac{6}{7} p_j(i) - \frac{6}{7} p_i(j) \cdot \frac{6}{7} p_j(i) \\
&\geq \frac{6}{7} \left( p_i(j) + p_j(i) - p_i(j) \cdot p_j(i) \right) \\
&= \frac{6}{7} \cdot p(i,j),
\end{aligned}
$$

which completes the proof. $\qquad\square$

We are now ready to prove Theorem 2. It is important to note that, although some of the proofs below are parallel to that of [27], our analysis needs to carefully take into account the approximation ratios introduced by the approximate KDE algorithm, which makes our analysis more involved. The following concentration inequalities will be used in our proof.

**Lemma 3** (Bernstein's Inequality [8]). *Let $X_1, \ldots, X_n$ be independent random variables such that $|X_i| \leq M$ for any $i \in \{1, \ldots, n\}$. Let $X = \sum_{i=1}^{n} X_i$, and $R = \sum_{i=1}^{n} \mathbb{E}\left[X_i^2\right]$. Then, it holds that*

$$\mathbb{P}\left[|X - \mathbb{E}[X]| \geq t\right] \leq 2 \exp\left(-\frac{t^2}{2(R + Mt/3)}\right).$$

**Lemma 4** (Matrix Chernoff Bound [28]). *Consider a finite sequence $\{X_i\}$ of independent, random, PSD matrices of dimension $d$ that satisfy $\|X_i\| \leq R$. Let $\mu_{\min} \triangleq \lambda_{\min}(\mathbb{E}[\sum_i X_i])$ and $\mu_{\max} \triangleq \lambda_{\max}(\mathbb{E}[\sum_i X_i])$. Then, it holds that*

$$\mathbb{P}\left[\lambda_{\min}\left(\sum_i X_i\right) \leq (1-\delta)\mu_{\min}\right] \leq d\left(\frac{e^{-\delta}}{(1-\delta)^{1-\delta}}\right)^{\mu_{\min}/R}$$

*for $\delta \in [0, 1]$, and*

$$\mathbb{P}\left[\lambda_{\max}\left(\sum_i X_i\right) \geq (1+\delta)\mu_{\max}\right] \leq d\left(\frac{e^{\delta}}{(1+\delta)^{1+\delta}}\right)^{\mu_{\max}/R}$$

*forr $\delta \geq 0$.*

*Proof of Theorem 2.* We first show that the degrees of all the vertices in the similarity graph $\mathsf{K}$ are preserved with high probability in the sparsifier $\mathsf{G}$. For any vertex $v_i$, let $y_{i,1}, \ldots, y_{i,L}$ be the indices of the neighbours of $v_i$ sampled by Algorithm 2.

For every pair of indices $i \neq j$, and for every $1 \leq l \leq L$, we define the random variable $Z_{i,j,l}$ to be the weight of the sampled edge if $j$ is the neighbour sampled from $i$ at iteration $l$, and 0 otherwise:

$$Z_{i,j,l} \triangleq \begin{cases} \frac{k(x_i, x_j)}{\widehat{p}(i,j)} & \text{if } y_{i,l} = j \\ 0 & \text{otherwise.} \end{cases}$$

Then, fixing an arbitrary vertex $x_i$, we can write

$$\deg_{\mathsf{G}}(v_i) = \sum_{l=1}^{L} \sum_{i \neq j} Z_{i,j,l} + Z_{j,i,l}.$$

We have

$$\mathbb{E}\left[\deg_{\mathsf{G}}(v_i)\right] = \sum_{l=1}^{L} \sum_{j \neq i} \mathbb{E}\left[Z_{i,j,l}\right] + \mathbb{E}\left[Z_{j,i,l}\right]$$

$$= \sum_{l=1}^{L} \sum_{j \neq i} \left[\mathbb{P}\left[y_{i,l} = j\right] \cdot \frac{k(x_i, x_j)}{\widehat{p}(i,j)} + \mathbb{P}\left[y_{j,l} = i\right] \cdot \frac{k(x_i, x_j)}{\widehat{p}(i,j)}\right].$$

By Lemmas 1 and 2 and Corollary 1, we have

$$\mathbb{E}\left[\deg_{\mathsf{G}}(v_i)\right] \geq \sum_{j \neq i} \frac{k(x_i, x_j)}{\widehat{p}(i,j)} \left(\frac{L \cdot k(x_i, x_j)}{2 \deg_{\mathsf{K}}(v_i)} + \frac{L \cdot k(x_i, x_j)}{2 \deg_{\mathsf{K}}(v_j)}\right)$$

$$= \sum_{i \neq j} \frac{3k(x_i, x_j)}{\widehat{p}(i,j)} \left(p_i(j) + p_j(i)\right)$$

$$\geq \sum_{i \neq j} \frac{5k(x_i, x_j)}{12} = \frac{5 \deg_{\mathsf{K}}(v_i)}{12},$$

where the last inequality follows by the fact that $\widehat{p}(i,j) \leq (36/5) \cdot p(i,j) \leq (36/5) \cdot (p_i(j) + p_j(i))$.
Similarly, we have

$$\mathbb{E}\left[\deg_{\mathsf{G}}(v_i)\right] \leq \sum_{j \neq i} \frac{k(x_i, x_j)}{\widehat{p}(i,j)} \left(\frac{2 \cdot L \cdot k(x_i, x_j)}{\deg_{\mathsf{K}}(v_i)} + \frac{2 \cdot L \cdot k(x_i, x_j)}{\deg_{\mathsf{K}}(v_j)}\right)$$

$$= \sum_{j \neq i} \frac{12 \cdot k(x_i, x_j)}{\widehat{p}(i,j)} \left(p_i(j) + p_j(i)\right)$$

$$\leq \sum_{j \neq i} 28 \cdot k(x_i, x_j) = 28 \cdot \deg_{\mathsf{K}}(v_i),$$

where the inequality follows by the fact that

$$\widehat{p}(i,j) \geq \frac{6}{7} \cdot p(i,j) = \frac{6}{7} \cdot (p_i(j) + p_j(i) - p_i(j) \cdot p_j(i)) \geq \frac{3}{7} \cdot (p_i(j) + p_j(i)).$$

In order to prove a concentration bound on this degree estimate, we would like to apply the Bernstein inequality for which we need to bound

$$R = \sum_{l=1}^{L} \sum_{j \neq i} \mathbb{E}\left[Z_{i,j,l}^2\right] + \mathbb{E}\left[Z_{j,i,l}^2\right]$$

$$= \sum_{l=1}^{L} \sum_{j \neq i} \mathbb{P}\left[y_{i,l} = j\right] \cdot \frac{k(x_i, x_j)^2}{\widehat{p}(i,j)^2} + \mathbb{P}\left[y_{j,l} = i\right] \cdot \frac{k(x_i, x_j)^2}{\widehat{p}(i,j)^2}$$

$$\leq \sum_{j \neq i} \frac{12 \cdot k(x_i, x_j)^2}{\widehat{p}(i,j)^2} \cdot (p_i(j) + p_j(i))$$

$$\leq \sum_{j \neq i} 28 \cdot \frac{k(x_i, x_j)^2}{\widehat{p}(i,j)}$$

$$\leq \sum_{j \neq i} \frac{98}{3} \cdot \frac{k(x_i, x_j)^2}{p_i(j)}$$

$$= \sum_{j \neq i} \frac{98}{3} \cdot \frac{k(x_i, x_j) \cdot \deg_{\mathsf{K}}(v_i) \cdot \lambda_{k+1}}{C \log(n)}$$

$$= \frac{98 \cdot \deg_{\mathsf{K}}(v_i)^2 \cdot \lambda_{k+1}}{3 \cdot C \log(n)},$$

where the third inequality follows by the fact that

$$\widehat{p}(i,j) \geq \frac{6}{7} \cdot p(i,j) \geq \frac{6}{7} \cdot p_i(j).$$

Then, by applying Bernstein's inequality we have for any constant $C_2$ that

$$\mathbb{P}\left[|\deg_{\mathsf{G}}(v_i) - \mathbb{E}[\deg_{\mathsf{G}}(v_i)]| \geq \frac{1}{C_2} \deg_{\mathsf{K}}(v_i)\right] \leq 2 \exp\left(-\frac{\deg_{\mathsf{K}}(v_i)^2/(2 \cdot C_2^2)}{\frac{98 \deg_{\mathsf{K}}(v_i)^2 \lambda_{k+1}}{3C \log(n)} + \frac{7 \deg_{\mathsf{K}}(v_i)^2 \lambda_{k+1}}{6 C C_2 \cdot \log(n)}}\right)$$

$$\leq 2 \exp\left(-\frac{C \cdot \log(n)}{((196/3) \cdot C_2^2 + (7/3) \cdot C_2) \cdot \lambda_{k+1}}\right)$$

$$= o(1/n),$$

where we use the fact that

$$Z_{i,j,l} \leq \frac{7k(x_i, x_j)}{6p_i(j)} = \frac{7 \deg_{\mathsf{K}}(v_i) \cdot \lambda_{k+1}}{6C \cdot \log(n)}.$$

Therefore, by taking $C$ to be sufficiently large and by the union bound, it holds with high probability that the degree of all the nodes in $\mathsf{G}$ are preserved up to a constant factor. For the remainder of the

proof, we assume that this is the case. Note in particular that this implies $\text{vol}_\mathsf{G}(S) = \Theta(\text{vol}_\mathsf{K}(S))$ for any subset $S \subseteq V$.

Next, we prove it holds for $\mathsf{G}$ that $\phi_\mathsf{G}(S_i) = O\left(k \cdot \phi_\mathsf{K}(S_i)\right)$ for any $1 \le i \le k$, where $S_1, \ldots, S_k$ form an optimal clustering in $\mathsf{K}$.

By the definition of $Z_{i,j,l}$, it holds for any $1 \le i \le k$ that

$$\mathbb{E}\left[\partial_\mathsf{G}(S_i)\right] = \mathbb{E}\left[\sum_{a \in S_i} \sum_{b \notin S_i} \sum_{l=1}^{L} Z_{a,b,l} + Z_{b,a,l}\right]$$
$$\le \sum_{a \in S_i} \sum_{b \notin S_i} \frac{12k(x_a, x_b)}{\widehat{p}(a,b)} \cdot (p_a(b) + p_b(a))$$
$$= O\left(\partial_\mathsf{K}(S_i)\right)$$

where the last line follows by Corollary 1. By Markov's inequality and the union bound, with constant probability it holds for all $i = 1, \ldots, k$ that

$$\partial_\mathsf{G}(S_i) = O(k \cdot \partial_\mathsf{K}(S_i)).$$

Therefore, it holds with constant probability that

$$\rho_\mathsf{G}(k) \le \max_{1 \le i \le k} \phi_\mathsf{G}(S_i) = \max_{1 \le i \le k} O(k \cdot \phi_\mathsf{K}(S_i)) = O(k \cdot \rho_\mathsf{K}(k)).$$

Finally, we prove that $\lambda_{k+1}(\mathbf{N}_\mathsf{G}) = \Omega(\lambda_{k+1}(\mathbf{N}_\mathsf{K}))$. Let $\overline{\mathbf{N}}_\mathsf{K}$ be the projection of $\mathbf{N}_\mathsf{K}$ on its top $n - k$ eigenspaces, and notice that $\overline{\mathbf{N}}_\mathsf{K}$ can be written

$$\overline{\mathbf{N}}_\mathsf{K} = \sum_{i=k+1}^{n} \lambda_i(\mathbf{N}_\mathsf{K}) f_i f_i^\mathsf{T}$$

where $f_1, \ldots, f_n$ are the eigenvectors of $\mathbf{N}_\mathsf{K}$. Let $\overline{\mathbf{N}}_\mathsf{K}^{-1/2}$ be the square root of the pseudoinverse of $\overline{\mathbf{N}}_\mathsf{K}$.

We prove that the top $n - k$ eigenvalues of $\mathbf{N}_\mathsf{K}$ are preserved, which implies that $\lambda_{k+1}(\mathbf{N}_\mathsf{G}) = \Theta(\lambda_{k+1}(\mathbf{N}_\mathsf{K}))$. To prove this, for each data point $x_i$ and sample $1 \le l \le L$, we define a random matrix $X_{i,l} \in \mathbb{R}^{n \times n}$ by

$$X_{i,l} = w_\mathsf{G}(v_i, v_j) \cdot \overline{\mathbf{N}}_\mathsf{K}^{-1/2} b_e b_e^\mathsf{T} \overline{\mathbf{N}}_\mathsf{K}^{-1/2};$$

where $j = y_{i,l}$, $b_e \triangleq \chi_{v_i} - \chi_{v_j}$ is the edge indicator vector, and $x_{v_i} \in \mathbb{R}^n$ is defined by

$$\chi_{v_i}(a) \triangleq \begin{cases} \frac{1}{\sqrt{\deg_\mathsf{K}(v_i)}} & \text{if } a = i \\ 0 & \text{otherwise.} \end{cases}$$

Notice that

$$\sum_{i=1}^{n} \sum_{l=1}^{L} X_{i,l} = \sum_{\text{sampled edges } e = \{v_i, v_j\}} w_\mathsf{G}(v_i, v_j) \cdot \overline{\mathbf{N}}_\mathsf{K}^{-1/2} b_e b_e^\mathsf{T} \overline{\mathbf{N}}_\mathsf{K}^{-1/2} = \overline{\mathbf{N}}_\mathsf{K}^{-1/2} \mathbf{N}_\mathsf{G}' \overline{\mathbf{N}}_\mathsf{K}^{-1/2}$$

where

$$\mathbf{N}_\mathsf{G}' = \sum_{\text{sampled edges } e = \{v_i, v_j\}} w_\mathsf{G}(v_i, v_j) \cdot b_e b_e^\mathsf{T}$$

is the Laplacian matrix of $\mathsf{G}$ normalised with respect to the degrees of the nodes in $\mathsf{K}$. We prove that, with high probability, the top $n - k$ eigenvectors of $\mathbf{N}_\mathsf{G}'$ and $\mathbf{N}_\mathsf{K}$ are approximately the same. Then, we show the same for $\mathbf{N}_\mathsf{G}$ and $\mathbf{N}_\mathsf{G}'$ which implies that $\lambda_{k+1}(\mathbf{N}_\mathsf{G}) = \Omega(\lambda_{k+1}(\mathbf{N}_\mathsf{K}))$.

We begin by looking at the first moment of $\sum_{i=1}^{n} \sum_{l=1}^{L} X_{i,l}$, and have that

$$\lambda_{\min} \left( \mathbb{E} \left[ \sum_{i=1}^{n} \sum_{l=1}^{L} X_{i,l} \right] \right) = \lambda_{\min} \left( \sum_{i=1}^{n} \sum_{l=1}^{L} \sum_{\substack{j \neq i \\ e = \{v_i, v_j\}}} \mathbb{P}\left[y_{i,l} = j\right] \cdot \frac{k(x_i, x_j)}{\widehat{p}(i,j)} \cdot \overline{\mathbf{N}}_{\mathsf{K}}^{-1/2} b_e b_e^{\intercal} \overline{\mathbf{N}}_{\mathsf{K}}^{-1/2} \right)$$

$$\geq \lambda_{\min} \left( \sum_{i=1}^{n} \sum_{\substack{j \neq i \\ e = \{v_i, v_j\}}} 3 p_i(j) \cdot \frac{k(x_i, x_j)}{\widehat{p}(i,j)} \cdot \overline{\mathbf{N}}_{\mathsf{K}}^{-1/2} b_e b_e^{\intercal} \overline{\mathbf{N}}_{\mathsf{K}}^{-1/2} \right)$$

$$\geq \lambda_{\min} \left( \frac{5}{12} \cdot \overline{\mathbf{N}}_{\mathsf{K}}^{-1/2} \mathbf{N}_{\mathsf{K}} \overline{\mathbf{N}}_{\mathsf{K}}^{-1/2} \right) = \frac{5}{12},$$

where the last inequality follows by the fact that

$$\widehat{p}(i,j) \leq \frac{36}{5} \cdot p(i,j) \leq \frac{36}{5} \cdot (p_i(j) + p_j(i)).$$

Similarly,

$$\lambda_{\max} \left( \mathbb{E} \left[ \sum_{i=1}^{n} \sum_{l=1}^{L} X_{i,l} \right] \right) = \lambda_{\max} \left( \sum_{i=1}^{n} \sum_{l=1}^{L} \sum_{\substack{j \neq i \\ e = \{v_i, v_j\}}} \mathbb{P}\left[y_{i,l} = j\right] \cdot \frac{k(x_i, x_j)}{\widehat{p}(i,j)} \cdot \overline{\mathbf{N}}_{\mathsf{K}}^{-1/2} b_e b_e^{\intercal} \overline{\mathbf{N}}_{\mathsf{K}}^{-1/2} \right)$$

$$\leq \lambda_{\max} \left( \sum_{i=1}^{n} \sum_{\substack{j \neq i \\ e = \{v_i, v_j\}}} 12 \cdot p_i(j) \cdot \frac{k(x_i, x_j)}{\widehat{p}(i,j)} \cdot \overline{\mathbf{N}}_{\mathsf{K}}^{-1/2} b_e b_e^{\intercal} \overline{\mathbf{N}}_{\mathsf{K}}^{-1/2} \right)$$

$$\leq \lambda_{\max} \left( 28 \cdot \overline{\mathbf{N}}_{\mathsf{K}}^{-1/2} \mathbf{N}_{\mathsf{K}} \overline{\mathbf{N}}_{\mathsf{K}}^{-1/2} \right) = 28,$$

where the last inequality follows by the fact that

$$\widehat{p}(i,j) \geq \frac{6}{7} \cdot p(i,j) \geq \frac{3}{7} \cdot (p_i(j) + p_j(i)).$$

Additionally, for any $i$ and $j = y_{i,l}$ and $e = \{v_i, v_j\}$, we have that

$$\|X_{i,l}\| \leq w_{\mathsf{G}}(v_i, v_j) \cdot b_e^{\intercal} \overline{\mathbf{N}}_{\mathsf{K}}^{-1/2} \overline{\mathbf{N}}_{\mathsf{K}}^{-1/2} b_e$$

$$= \frac{k(x_i, x_j)}{\widehat{p}(i,j)} \cdot b_e^{\intercal} \overline{\mathbf{N}}_{\mathsf{K}}^{-1} b_e$$

$$\leq \frac{k(x_i, x_j)}{\widehat{p}(i,j)} \cdot \frac{1}{\lambda_{k+1}} \|b_e\|^2$$

$$\leq \frac{7 \cdot \lambda_{k+1}}{3C \log(n) \left( \frac{1}{\deg_{\mathsf{K}}(v_i)} + \frac{1}{\deg_{\mathsf{K}}(v_j)} \right)} \cdot \frac{1}{\lambda_{k+1}} \left( \frac{1}{\deg_{\mathsf{K}}(v_i)} + \frac{1}{\deg_{\mathsf{K}}(v_j)} \right)$$

$$\leq \frac{7}{3C \log(n)}.$$

Now, we apply the matrix Chernoff bound and have that

$$\mathbb{P}\left[\lambda_{\max} \left( \sum_{i=1}^{n} \sum_{l=1}^{L} X_{i,l} \right) \geq 42 \right] \leq n \left( \frac{e^{1/2}}{(1 + 1/2)^{3/2}} \right)^{12C \cdot \log(n)} = O(1/n^c)$$

for some constant $c$. The other side of the matrix Chernoff bound gives us that

$$\mathbb{P}\left[\lambda_{\min} \left( \sum_{i=1}^{n} \sum_{l=1}^{L} X_{i,l} \right) \leq 5/24 \right] \leq O(1/n^c).$$

Combining these, with probability $1 - O(1/n^c)$ it holds for any non-zero $x \in \mathbb{R}^n$ in the space spanned by $f_{k+1}, \ldots, f_n$ that

$$\frac{x^\intercal \overline{\mathbf{N}}_{\mathsf{K}}^{-1/2} \mathbf{N}_{\mathsf{G}}' \overline{\mathbf{N}}_{\mathsf{K}}^{-1/2} x}{x^\intercal x} \in (5/24, 42).$$

By setting $y = \overline{\mathbf{N}}_{\mathsf{K}}^{-1/2} x$, we can rewrite this as

$$\frac{y^\intercal \mathbf{N}_{\mathsf{G}}' y}{y^\intercal \overline{\mathbf{N}}_{\mathsf{K}}^{1/2} \overline{\mathbf{N}}_{\mathsf{K}}^{1/2} y} = \frac{y^\intercal \mathbf{N}_{\mathsf{G}}' y}{y^\intercal \overline{\mathbf{N}}_{\mathsf{K}} y} = \frac{y^\intercal \mathbf{N}_{\mathsf{G}}' y}{y^\intercal y} \frac{y^\intercal y}{y^\intercal \overline{\mathbf{N}}_{\mathsf{K}} y} \in (5/24, 42).$$

Since $\dim(\mathrm{span}\{f_{k+1}, \ldots, f_n\}) = n - k$, we have shown that there exist $n - k$ orthogonal vectors whose Rayleigh quotient with respect to $\mathbf{N}_{\mathsf{G}}'$ is $\Omega(\lambda_{k+1}(\mathbf{N}_{\mathsf{K}}))$. By the Courant-Fischer Theorem, we have $\lambda_{k+1}(\mathbf{N}_{\mathsf{G}}') = \Omega(\lambda_{k+1}(\mathbf{N}_{\mathsf{K}}))$.

It only remains to show that $\lambda_{k+1}(\mathbf{N}_{\mathsf{G}}) = \Omega(\lambda_{k+1}(\mathbf{N}_{\mathsf{G}}'))$, which implies that $\lambda_{k+1}(\mathbf{N}_{\mathsf{G}}) = \Omega(\lambda_{k+1}(\mathbf{N}_{\mathsf{K}}))$. By the definition of $\mathbf{N}_{\mathsf{G}}'$, we have that $\mathbf{N}_{\mathsf{G}} = \mathbf{D}_{\mathsf{G}}^{-1/2} \mathbf{D}_{\mathsf{K}}^{1/2} \mathbf{N}_{\mathsf{G}}' \mathbf{D}_{\mathsf{K}}^{1/2} \mathbf{D}_{\mathsf{G}}^{-1/2}$. Therefore, for any $x \in \mathbb{R}^n$ and $y = \mathbf{D}_{\mathsf{K}}^{1/2} \mathbf{D}_{\mathsf{G}}^{-1/2} x$, it holds that

$$\frac{x^\intercal \mathbf{N}_{\mathsf{G}} x}{x^\intercal x} = \frac{y^\intercal \mathbf{N}_{\mathsf{G}}' y}{x^\intercal x} = \Omega\left(\frac{y^\intercal \mathbf{N}_{\mathsf{G}}' y}{y^\intercal y}\right),$$

where the final guarantee follows from the fact that the degrees in $\mathsf{G}$ are preserved up to a constant factor. The conclusion of the theorem follows by the Courant-Fischer Theorem.

Finally, we bound the running time of Algorithm 2 which is dominated by the recursive calls to Algorithm 1. We note that, although the number of nodes doubles at each level of the recursion tree (visualised in Figure 4), the total number of samples $S$ and data points $X$ remain constant for each level of the tree. Then, since the running time of the $\mathsf{KDE}$ algorithm is superadditive, the total running time of the $\mathsf{KDE}$ algorithms at level $i$ of the tree is

$$T_i = \sum_{j=1}^{2^i} T_{\mathsf{KDE}}(|S_{i,j}|, |X_{i,j}|, \epsilon)$$

$$\leq T_{\mathsf{KDE}}\left(\sum_{j=1}^{2^i} |S_{i,j}|, \sum_{j=1}^{2^i} |X_{i,j}|, \epsilon\right) = T_{\mathsf{KDE}}(|S|, |X|, \epsilon).$$

Since there are $O(\log(n))$ levels of the tree, the total running time of Algorithm 1 is $\widetilde{O}(T_{\mathsf{KDE}}(|S|, |X|, \epsilon))$. This completes the proof. $\qquad\square$

## B Additional Experimental Results

In this section, we include in Figures 6 and 7 some additional examples of the performance of the six spectral clustering algorithms on the BSDS image segmentation dataset. Due to its quadratic memory requirement, the SKLEARN GK algorithm cannot be used on the full-resolution image. Therefore, we present its results on each image downsampled to 20,000 pixels. For every other algorithm, we show the results on the full-resolution image. In every case, we find that our algorithm is able to identify more refined detail of the image when compared with the alternative algorithms.

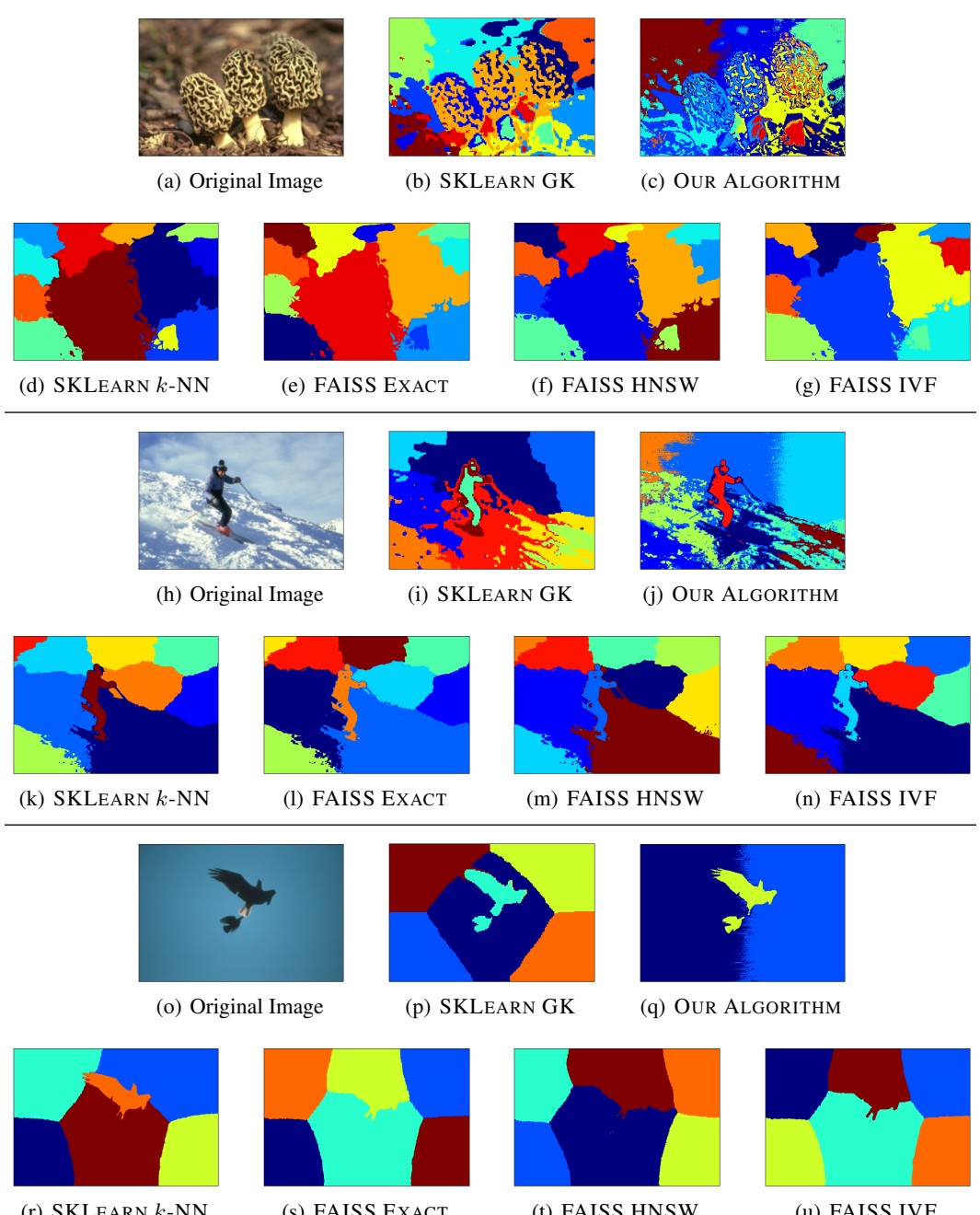

(a) Original Image     (b) SKLearn GK     (c) Our Algorithm

(d) SKLearn $k$-NN     (e) FAISS Exact     (f) FAISS HNSW     (g) FAISS IVF

(h) Original Image     (i) SKLearn GK     (j) Our Algorithm

(k) SKLearn $k$-NN     (l) FAISS Exact     (m) FAISS HNSW     (n) FAISS IVF

(o) Original Image     (p) SKLearn GK     (q) Our Algorithm

(r) SKLearn $k$-NN     (s) FAISS Exact     (t) FAISS HNSW     (u) FAISS IVF

Figure 6: More examples on the performance of the spectral clustering algorithms for image segmentation.

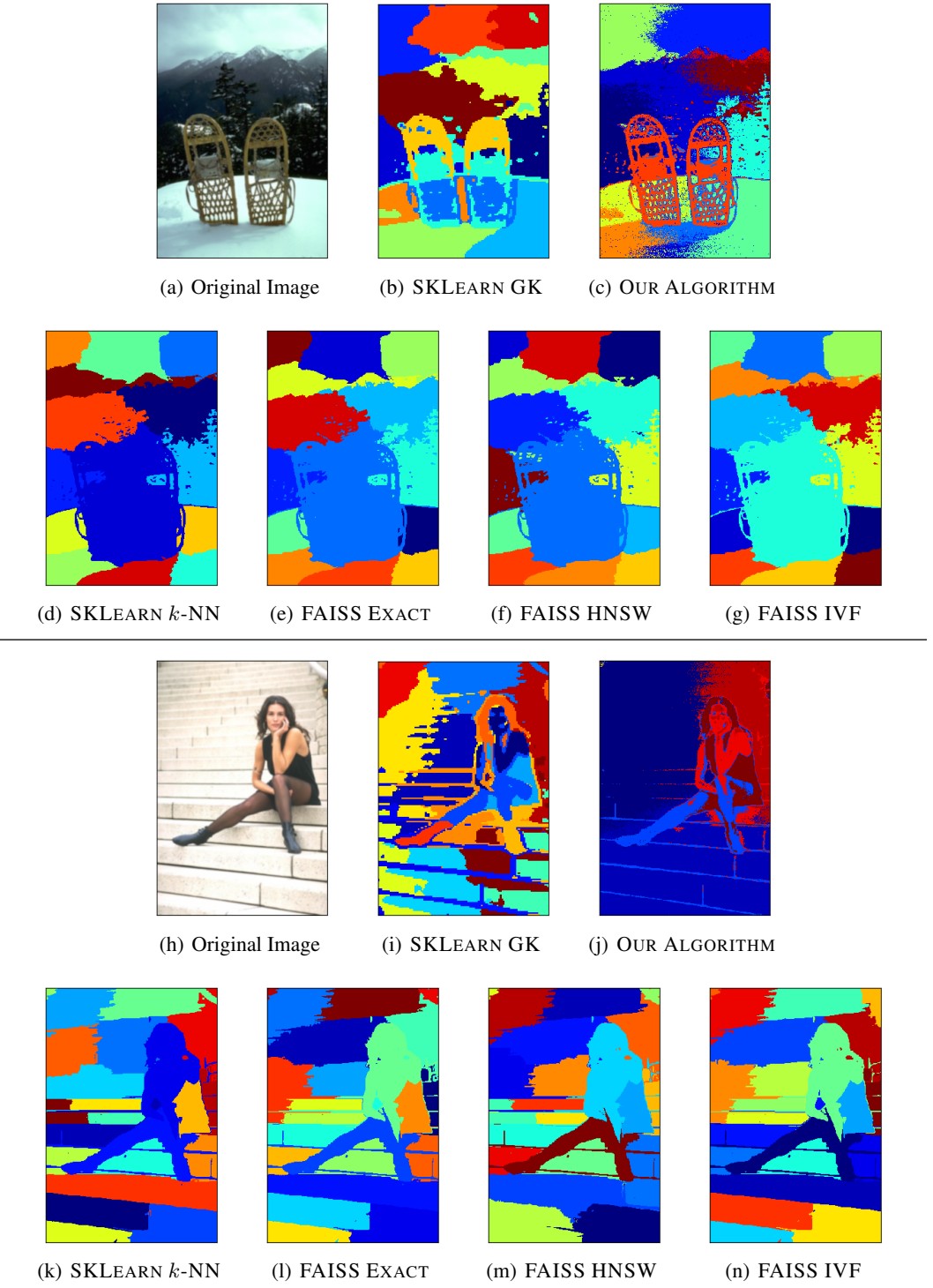

(a) Original Image  (b) SKLEARN GK  (c) OUR ALGORITHM

(d) SKLEARN $k$-NN  (e) FAISS EXACT  (f) FAISS HNSW  (g) FAISS IVF

(h) Original Image  (i) SKLEARN GK  (j) OUR ALGORITHM

(k) SKLEARN $k$-NN  (l) FAISS EXACT  (m) FAISS HNSW  (n) FAISS IVF

Figure 7: More examples on the performance of the spectral clustering algorithms for image segmentation.

