# OpenReview forum: "Fast Approximation of Similarity Graphs with Kernel Density Estimation"
_NeurIPS.cc/2023/Conference — NeurIPS 2023 spotlight_

### Official Review · Reviewer_oQAs · 2023-06-19

**Soundness:** 3 good
**Presentation:** 3 good
**Contribution:** 2 fair
**Rating:** 6
**Confidence:** 4

**Summary:**

Given a set of n points and a pairwise kernel k(x,y), the fully connected similarity graph K is the weighted graph that has an edge between every pair x,y with weight k(x,y). The paper presents an algorithm for computing a sparse graph G that approximates K in a certain sense that is useful for spectral clustering. The algorithm is based on black-box calls to an efficient kernel density estimation (KDE) algorithm. For some kernels such black-boxes are available and have been the focus of much work recently, and they lead to efficient graph construction in this paper. The result is given in the form of a formal theorem and empirical evaluation.


**Strengths:**

The paper is generally clearly written and contains nice and convincing theoretical and empirical findings that contribute to the literature on the topic and could be useful in practice. It is a nice application of the recent progress on efficient KDE to graph clustering. Even though it may not be as novel and general and advertised (see below), I still find it above the bar for acceptance.

**Weaknesses:**

(1) Novelty is somewhat limited, and seems overstated. The intro (lines 61-62) claims "a novel connection between the KDE and the fast construction of similarity graphs". This connection is quite well-known already and has been the focus of some recent works, including [1], [6] and "Spectral Sparsification of Metrics and Kernels" (Quanrud, SODA 2021). Anyway this connection between efficient KDE and kernel similarity graphs is very natural and straightforward, and it is unsurprising that all these works appeared shortly after the recent progress on efficient KDE. These works instantiate the connection in possibly different ways, but the general idea is similar. The novelty in this work seems to me restricted to the specific technical application to the SZ clustering framework, and not so much a conceptual point about kernels and graphs.

(2) A more specific concern about novelty is that the main sampling method at the heart of the algorithm (lines 195-204, Algorithm 1) seems to have appeared already in [6] (see their "sample random neighbor" primitive). The work is cited only for a certain conditional hardness result, while its very similar algorithmic ideas are not mentioned. How do these techniques relate to each other?

(3) The generality of the result itself is also overstated, particularly in the abstract that the result is "applicable to arbitrary kernels". I suppose this alludes to the fact that the algorithm uses KDE as a black-box, this is rather misleading, since there relatively few and specialized instances of known black-boxes of the type needed for this algorithm, and for a relatively limited set of kernels. For arbitrary kernels the black-box does not exist and the result of the paper is not applicable.

This affects the writing in the intro. Lines 35-40 formulate the main question as "constructing a sparse graph that preserves the cluster structure", focusing on sparsity and eschewing running time, possibly in order to circumvent said limitations on generality. However, this renders the main question moot, since trivially the full graph can be constructed and then spectrally sparsified. It would probably have been preferable to promptly discuss running times with their inevitable limitations.

**Questions:**

1. Please clarify connection to sampling in [6] (see above)

2. The paragraph about LSH in lines 90-94 (related work) sounds odd. It says it's unclear how to use LSH for approximating geometric similarity graphs. However, much of the work you cite for efficient KDE builds directly on LSH (and consequently, so do some instantiations of your algorithm, as well as similar graph algorithms from prior work mentioned earlier).

**Limitations:**

No major concerns, though some limitations on the generality of the results may not be sufficiently discussed, as mentioned above in weaknesses.

---

> ### Author Rebuttal · Authors · 2023-08-09
>
> Many thanks for your positive and detailed report, and valuable comments. We will take these into account when preparing the final version of our paper. Here are our responses to your questions and comments.
>
> **Question 1.**
> We agree that there is some similarity between our algorithm and the one in [6].
> We remark that our algorithm involves sampling a random neighbour of *every* vertex in a single procedure, whereas the one in [6] samples a neighbour of one vertex.
> Due to this difference, the analysis of our algorithm requires a careful consideration of each row of the recursion tree in Figure 4 with respect to the sets $S_{i, j}$ and $X_{i, j}$.
> Furthermore, our application of these samples to construct a cluster-preserving sparsifier is novel and requires a much more involved analysis than the analysis of SZ [21] due to the sampling method based on KDE.
>
> **Question 2.**
> We will revise this paragraph in the final version.
>
> **Weakness 1.**
> We agree that the novelty in our work is the application of efficient KDE to the construction of a cluster-preserving sparsifier of the complete kernel similarity graph.
> It was not our intention to claim more than this, and following your comment we will make it clearer in the final version.
>
> **Weakness 2.**
> See the answer to Question 1.
>
> **Weakness 3.**
> Based on our black-box reduction, our algorithm is applicable to any kernel with an efficient KDE algorithm.
> We will clarify this and remove the claim that our algorithm is applicable to arbitrary kernels.
> As you point out, our goal is to develop an algorithm with  fast running time for constructing a sparse similarity graph, and we will make this clearer in the introduction.

---

> > ### Comment · Reviewer_oQAs · 2023-08-14
> >
> > Thank you for your answers. About [6], I understand that your application of this sampling method (to many vertex simultaneously) and the analysis around it are different from [6], since you use it to a different end. But nonetheless, the sampling method that you present as the main idea in the beginning of section 4 is the same---calling it "some similarity" seems a gross understatement given how identical it is to [6]---and is presented without mention nor acknowledgement that it has appeared in prior work. This does not seem acceptable to me. Reiterating my original review, this submission is okay for acceptance in terms of its technical content, but the substantial issues with its claims to generality and novelty and lack of reference to prior work need to be addressed and not minimized or dismissed.

---

### Official Review · Reviewer_hhPL · 2023-07-01

**Soundness:** 4 excellent
**Presentation:** 2 fair
**Contribution:** 3 good
**Rating:** 6
**Confidence:** 4

**Summary:**

This work presents an algorithm framework for creating a (weighted) sparse similarity graph, which is an important object for a variety of ML problems. The algorithm samples pairs of points to include in the graph using a sampler based on kernel density estimation (KDE). The idea is to recursively split the data into subsets, compute the KDE on each subset, and select a subset proportional to the density. This process is done for every point in the dataset (node in the graph), resulting in a collection of edges in the sparse graph.

A nice feature of this method is that it provably preserves the spectral clustering properties of the graph. To my knowledge, this is the only sub-quadratic (in space / time) algorithm that does this. The method is also agnostic to the choice of KDE solver, leading to interesting practical algorithms. When paired with a classical KDE method (the Fast Gauss Transform), the method outperforms a number of reasonable baselines for graph construction + spectral clustering in low dimensions.

**Strengths:**

**S1: Novel and interesting reduction,** from KDE to similarity graph construction. The sampling mechanism (Algorithm 1) is particularly nice, and it likely has further applications to other settings where we want to sample proportional to kernel sums. The proposed framework also benefits from and motivates the growing body of work on the KDE problem. This leads to exciting possibilities for practical algorithms, depending on which KDE method is used in the framework.

**S2: Good theoretical results.** It is great to see rigorous guarantees on the quality of the subsequent partitioning of the similarity graph as well as the time / space complexity. Many studies only perform the latter, as it is difficult to prove this kind of result.

**S3: Well-engineered implementation,** with reproducible experiments. I was able to fully reproduce all of the experiments in this paper. I think the implementation will be of use to the community.

**Weaknesses:**

**W1: Notation.** The notation in Section 2 is understandable but would benefit from some polishing. For example, $w_G(u, v)$ is the edge weight function over node inputs and is defined only for those $(u,v)$ that are in the graph. But there is a $w_G(S, V\setminus S)$ (with a different-script $G$ and set-of-node inputs), which is the sum of outbound edge weights from within $S$ to other parts of the graph. There is also a $w(u,v)$ used in the definition of the node degree, which isn't defined anywhere. A graph is defined as $G = (V, E)$ (with a separately-specified weight function) but the similarity graph is given as $K = (V, E, \mathrm{weights})$. The kernel density sum KDE is introduced with a subscript $g_{[a,b]}$, but the analysis doesn't use this notation and later introduces $g_X$ as the KDE sum over a set of points $X$. This is all a bit confusing.

One possible fix is to define everything in terms of a weighted graph $G = (V, E, w)$. Here, $w$ is the weighting function from $(u,v)\to \mathbb{R}^{+}$ where $w(u,v) = $ edge weight if $(u,v) \in G$ and 0 otherwise. This would give a much simpler definition for the adjacency matrix ($A_{ij} = w(v_i, v_j)$), the degree ($d(v) = \sum_{u \in V} w(v, u)$), and the outbound edge weight sum, which can be written as a unary function $w_o(S) = \sum_{u \in S, v \not \in S} w(u,v)$ to further reduce confusion with $w(u,v)$. Also meanings of $\rho_K$ and $\lambda$ should also be introduced before they are used in the main theorem, even if the full definition is deferred until later.

**W2: Literature positioning / context.** The flexibility of the framework is a major strength of the paper, so it would be nice to highlight the (growing) number of KDE algorithms. For example:
- Hashing-Based Estimators (HBE): Uses LSH tables to do KDE, where we only compute the KDE over the points that collide in the LSH table. The original paper was in [[FOCS 2017]](https://arxiv.org/abs/1808.10530), with a practical version in [[ICML 2019]](http://proceedings.mlr.press/v97/siminelakis19a) and some incremental progress in [[NeurIPS 2019]](https://openreview.net/forum?id=H1xEABrgIH). (only provably works for some kernels, fast in high dimensions, good in practice).
- Interpolation KDE: [[AISTAT 2021]](http://proceedings.mlr.press/v130/turner21a/turner21a.pdf) performs KDE via polynomial interpolation (for arbitrary smooth kernels).
- RACE: [[WWW 2020]](https://dl.acm.org/doi/fullHtml/10.1145/3366423.3380244) performs fast, online KDE, using a few MB of space (only for a limited set of kernels, but extremely space / time efficient).
- Discrepancy coresets: [[COLT 2019]](https://proceedings.mlr.press/v99/karnin19a.html) performs KDE using online coresets (a re-weighted sub-sample of the dataset). In practice, this requires slower preprocessing than other methods but is more general / can be more accurate.

Each of these methods will present unique tradeoffs when incorporated into this framework. While a full comparison of various KDE subroutines is probably out-of-scope, it would be nice to discuss some of these tradeoffs / make recommendations. A few similarity graph construction methods have also been developed since the review paper by Luxburg in 2007, which should be mentioned:

- NN Descent: [[WWW 2011]](https://dl.acm.org/doi/abs/10.1145/1963405.1963487) This is an iterative technique that progressively refines a random guess for the initial k-NN graph.
- FLASH: [[SIGMOD 2018]](https://dl.acm.org/doi/abs/10.1145/3183713.3196925) This is an LSH-based method that constructs a similarity graph (i.e. finds all pairs $(u,v)$ such that $w(u,v)$ exceeds a threshold). This is much faster than other LSH methods because it uses clever counting tricks to avoid doing any explicit distance / kernel calculations.

Finally, the method in the paper, "Learning space partitions for nearest neighbor search" [[ICLR 2020]](https://openreview.net/forum?id=rkenmREFDr), might be relevant. This paper provides a framework for approximate near neighbor search that starts with a similarity graph, then partitions the data via a balanced min-cut partitioning of the graph. These partitions are then used to do a partition-based (FAISS IVF-style) similarity search, with provable guarantees (first theory for this type of algorithm). From a practical perspective, the hard part of this framework is to get the initial similarity graph, so your algorithm might help.


**W3: Experiments.** A couple of highly competitive baselines are not represented in the experimental evaluation, notably FLASH (which is $O(N \mathrm{polylog}(N))$) and NN-Descent (which scales about $O(N^{1.15})$, based on empirical results). For example, the C++ implementation of FLASH can construct a similarity graph for the webspam dataset (N = 350k) in under 10 seconds and for the friendster dataset (N = 65 million) in 1578 seconds. This 3-10x faster than the result of extrapolating Figure 5, and these datasets are of much higher dimension than two moons.

The datasets in the experiments are also all low-dimensional: even the BSDS task is only 5-dimensional. Many clustering applications (e.g. those that handle text embeddings) have inputs in the hundreds of dimensions. It would be nice to understand how this method performs in the higher-dimensional setting.

**Questions:**

1. Does T_KDE include preprocessing + query time? Most of the recent work in fast KDE requires a preprocessing algorithm (typically $O(N)$) followed by a query algorithm (typically much faster than $O(N)$). It looks like this is the case based on the T_KDE reported for FGT and the LSH method, but it wasn't clear from the definition. This is important because Algorithm 1 calls the KDE subroutine on different subsets of X.

2. What is the state of the art result (using e.g. neural network models) on the BSDS image segmentation dataset? Is it close to the result for k-NN / similarity graph algorithms or is it much better?

**Minor items:**
To get this to compile under MacOS arm64 (M1 chips), I did the following.
1. Pass an up-to-date clang compiler to cmake (via -D CMAKE_CXX_COMPILER).
2. Add the flag "-I/usr/local/include" to the compile_args in setup.py.
3. Break the function declarations in stag_lib/KMeansRex/mersenneTwister2002.c into a header.
4. pip install stag (stag isn't in requirements.txt, for some reason)

Nitpick on Line 92: "Despite extensive research on LSH-based algorithms, it remains unclear whether such techniques can be employed to approximately construct a similarity graph in nearly-linear time."
This has been done at scale (albeit without theoretical guarantees on the quality of the similarity graph), see section 4 of [the FLASH paper](https://arxiv.org/pdf/1709.01190.pdf).

**Limitations:**

I do not foresee any negative societal impacts of this work.

---

> ### Author Rebuttal · Authors · 2023-08-09
>
>  Many thanks for your positive and detailed report, and valuable comments. We will take these into account when preparing the final version of our paper. Here are our responses to your questions and comments.
>
> **Question 1: KDE preprocessing and query time.**
> In our paper, $T_{\mathrm{KDE}}$ includes both the preprocessing and query time of the KDE algorithm.
> As you correctly point out, our algorithm applies the KDE algorithm with  different subsets of the data points, and the preprocessing step is required each time.
> We will clarify this in the final version of the paper.
>
> **Question 2: State-of-the-art on the BSDS dataset.** The state-of-the-art for the (specific) BSDS dataset is recorded in [this paper](https://ieeexplore.ieee.org/document/5557884), and this achieves a Rand Index of 0.85. However, to the best of our knowledge, the Segment Anything Model (SAM) available [here](https://segment-anything.com/) presents the state-of-the-art for most image segmentation datasets. SAM has been trained on a dataset of 11 million images, and produces much better segmentation results than most unsupervised approaches. We emphasise that our algorithm is a general clustering algorithm, and is not developed specifically for image segmentation.
>
> **Weakness 1.**  We agree  with your comment. We'll take this into account, and improve our use of notation in the final version of the paper.
>
> **Weakness 2.** We're pleased to see that your consider the flexibility of our developed framework as a major strength of the paper. Following your suggestion, we will spend more effort discussing  recent work on KDE algorithms, and their tradeoffs when incorporated into our developed framework.
>
> **Weakness 3.** We'll discuss the application of our algorithm on higher-dimensional datasets, and compare it against the algorithms (FLASH and NN-Descent) that you mentioned in the final version. Conducting such experimental studies is difficult at the moment, due to the limited time of the rebuttal stage.
>
> **Minor items.**
> We are pleased that you were able to reproduce our experimental results with the provided code and for your comments on compiling and running the code.
> We will update our README based on your feedback.

---

> > ### Comment · Reviewer_hhPL · 2023-08-11
> >
> > Thanks for the response!
> >
> > I am still not fully convinced on the experiments, but I am willing to accept the segmentation results as evidence that the algorithm scales to N = 150k (though not to high dimensions). Today's most interesting applications of clustering (in NLP, vision, search, recommendation, etc) are in much higher dimensions, and it is not clear from these experiments whether this algorithm will yield SOTA or even scale all that well.
> >
> > However, I do not think this is a fatal issue. The main contribution of the paper lies in proposing a general framework, and the theoretical contribution is strong enough to place this paper above the bar for acceptance. Furthermore, the experimental evaluation is on just one instantiation of the framework - given the recent work in the KDE area, this is likely not even the strongest configuration, so the method might actually be a lot better in practice than it seems from these results. I'll be very interested to see the comparison with FLASH / NN-Descent (there are also likely to be some other, more recent similarity graph construction methods that would need to be considered if this were primarily an empirical paper). I do understand the difficulty of turning around experiments in the short rebuttal time frame, though, and issues with experiments should not count against the submission.

---

### Official Review · Reviewer_eUVC · 2023-07-06

**Soundness:** 3 good
**Presentation:** 4 excellent
**Contribution:** 4 excellent
**Rating:** 7
**Confidence:** 4

**Summary:**

The paper proposed an efficient approximation of the similarity graph using the Kernel Density Estimation (KDE) method as a black box. The proposed framework can preserve the clustering structure and can be run in nearly linear time. Lastly, experiments on synthetic and image segmentation datasets are provided to support these claims.


**Strengths:**

1. [Originality] The author proposed a novel algorithm by combining the ideas of subset/subspace dividing (somewhat related to LHS) with the help of probability measure estimated from KDE.
1. [Clarity] An excellent written paper that provides clear insights into motivations and formulation. Great overview of how the algorithm works Figures 3-4 and Algorithms 1-2. The fast runtime is achieved by the $O(\log(n))$ recursive call to the KDE as well as the superaddictive property of the $T_{KDE}$.
1. [Quality] The authors compared their proposed algorithm with publicly available nearest neighbor implementations (FAISS, sklearn) and show drastic speedup in the large sample size regime.
1. [Significance] I believe this paper makes a significant contribution in the unsupervised learning domains. The graph sparsification/Laplacian construction usually is done in a two-way approach, i.e., calculate pairwise distances and compute the similarity graph (with an additional sparsification step). The proposed method can do the aforementioned steps in a single step, thus drastically reducing the run time.


**Weaknesses:**

1. The authors have shown theoretically that the proposed algorithm is a {\em cluster-preserving sparsifier}. This claim can be further supported by showing a comparison between the eigenvalues (or eigengaps) of the true, graph sparsifiers (e.g., from [13, 14, 19]), and the proposed method.
1. There is another kind of similarity graph, called the $\varepsilon$-radius graph. The graph $G(V, E)$ is constructed with $E = { (x_i, x_j) \in V^2 :  \|x_i - x_j\| \leq \varepsilon}$. I think it will be informative to discuss this in the Introduction (L27-34).
1. It is quite confusing to see Theorem 1 without the full definition, or even a high-level description, of what $\rho$, $\lambda$, $N_K$ are in Section 1. Readers can somewhat guess what they are based on notational convention, but to improve readability, I would suggest to either include the full definition or (even better) simplify it to make it less rigorous when it is first introduced in Section 1 (then provided a detailed version later in Section 4 or Appendix if no space).
1. For the runtime of the FGT (L148), it might be more informative to write out all the terms inside the log rather than writing $\widetilde O(m+n)$, because the error term $\epsilon$ is hidden in the log.
1. It will be interesting to see experiments on higher dimensional (real or synthetic) datasets and compare the runtime correspondingly. The experiments are run on (relatively) low-dimensional and straightforward examples, e.g., synthetic datasets (d=2) and image data (d=5). Some potential high-dimensional real datasets to consider e.g., [A,B].

---
[A] Mahmoud, Eman, Ali Takey, and Amin Shoukry. “Spectral Clustering for Optical Confirmation and Redshift Estimation of X-Ray Selected Galaxy Cluster Candidates in the SDSS Stripe 82.” Astronomy and Computing 16 (2016): 174–84.

[B] Chmiela, Stefan, Alexandre Tkatchenko, Huziel E. Sauceda, Igor Poltavsky, Kristof T. Schütt, and Klaus-Robert Müller. “Machine Learning of Accurate Energy-Conserving Molecular Force Fields.” Science Advances 3, no. 5 (2017): e1603015.


**Questions:**

1. Related to Weakness #2, in practice, people usually use exponentially decaying kernel $k$ for the fully connected graph. You can approximate the fully connected graph with $\varepsilon$-radius graph by choosing the radius $\varepsilon$ and the bandwidth $\sigma$ appropriately, e.g., if using the exponential kernel as per L132, you can choose $\varepsilon = 3\cdot \sigma$; that way the far enough pairs will have small kernel values, i.e., $=\exp(-9) \sim 10^{-4}$, thus is negligible. I am curious if your method can be extended to the $\varepsilon$-radius graph under this scenario as well?
1. I am curious how difficult it is to extend this framework to the scenario of higher-order (neighborhood) simplicial complex and/or k-Laplacian. There are several works trying to use this information in data analysis [A-D], as well as algorithms [E] to sparsify the higher-order simplicial complexes with higher-order Cheerger constant. The reason I am asking is because the Higher-order k-Laplacian has dependency on $O(n_k)$ where $n_k$ is the cardinality of the k-complex (e.g., size of edges is $n_1 = |E|$), so if we can successfully sparsify the simplicial complex and/or k-Laplacian it can reduce the runtime drastically for those studies.
1. How can this framework be extended to the manifold learning or diffusion map setting? In this scenario, we care not only about the approximation of the null space but also the first few eigne-values. Can we have similar guarantees for these scenarios?
1. [Minor language usage] L21-23 (“Thanks to its out-performance over traditional clustering algorithms like k-means, this approach has…”) might be able to be improved by something like “Due to its superior performance compared to conventional clustering algorithms such as k-means, this approach has….”
1. [Minor notational consistency] For the equation between L194-195, you might be able to improve the clarity by changing the $z \in X_1$ to $x_j \in X_1$ and the $x_j$ in summation to another variable. This is because you used $x_i$ and $x_j$ in the discussion above (L192). Similar to L13 in Algorithm 1, you might want to reconsider the use of $y_i$ there to make the notation more consistent.

---
[A] Dey, Tamal K., Jian Sun, and Yusu Wang. “Approximating Loops in a Shortest Homology Basis from Point Data.” In Proceedings of the Twenty-Sixth Annual Symposium on Computational Geometry, 166–75, 2010.

[B] Chazal, Frédéric, Brittany Fasy, Fabrizio Lecci, Bertrand Michel, Alessandro Rinaldo, Alessandro Rinaldo, and Larry Wasserman. “Robust Topological Inference: Distance to a Measure and Kernel Distance.” The Journal of Machine Learning Research 18, no. 1 (2017): 5845–84.

[C] Chen, Yu-Chia, and Marina Meila. “The Decomposition of the Higher-Order Homology Embedding Constructed from the k-Laplacian.” Advances in Neural Information Processing Systems 34 (2021).

[D] Keros, Alexandros D., Vidit Nanda, and Kartic Subr. “Dist2cycle: A Simplicial Neural Network for Homology Localization.” In Proceedings of the AAAI Conference on Artificial Intelligence, 36:7133–42, 2022.

[E] Osting, Braxton, Sourabh Palande, and Bei Wang. “Spectral Sparsification of Simplicial Complexes for Clustering and Label Propagation.” ArXiv:1708.08436 [Cs], February 1, 2019. http://arxiv.org/abs/1708.08436.



**Limitations:**

I believe that the authors have covered all the limitations of the algorithms (e.g., it is only cluster-preserving not spectral sparsification, runtime will depend on the implementation of $T_KDE$, etc.) in the current version, but adding it and summarize it in a specific section will be beneficial.

Negative social impact is not applicable in this case, as this work primarily constitutes a theoretical contribution.

---

> ### Author Rebuttal · Authors · 2023-08-09
>
> Many thanks for your positive and detailed report, and valuable comments. We will take these into account when preparing the final version of our paper. Here are our responses to your questions and comments.
>
> **Question 1.**
> Our method constructs a sparsifier which has the same cluster structure as the fully-connected kernel graph.
> As you point out, for an exponentially-decaying kernel the $\epsilon$-radius graph will have a very similar structure to the kernel graph.
> Hence our constructed graph could also be viewed as a cluster-preserving sparsifier of an $\epsilon$-radius graph.
>
> **Questions 2 and 3.**
> Thank you for the questions on generalising our approach to higher-order structures, and to manifold learning.
> These are excellent directions for future work, but we have not considered these so far.
> We will add some discussion in the final version of the paper.
>
> **Questions 4 and 5.**
> We will make necessary changes based on your suggestions.
>
> **Weakness 1.** Classical algorithms for constructing spectral sparsifiers [13, 14, 19] are either based on a complicated graph decomposition framework, or Laplacian solvers. Despite of their nearly-linear time running time proven in theory, implementing these algorithms is a very challenging task. In fact, as far as we know, there is no publicly available implementation of nearly-linear time algorithms for spectral sparsifiers that work for a large-scale graph, and hence comparing the eigenvalues (or eigen-gap) between our constructed graph and spectral sparsifiers isn't feasible for a large-scale graph.
>
> Taking this into account, we compare the eigen-gap of our sparsifier with the ones of a fully connected graph, and an SZ sparsifier. We study the two moon dataset, and in Table 1 in the attached PDF we report the eigengap $\lambda_3 / \lambda_2$, where $\lambda_i$ is the $i$th smallest eigenvalue of the normalised Laplacian matrix of each constructed graph. We find that both the SZ algorithm and our proposed algorithm preserve a large eigen-gap between $\lambda_2$ and $\lambda_3$,  which implies that the cluster structure of the graph is preserved.
>
> **Weaknesses 2 - 4.**
> We will incorporate these editorial suggestions into the final version of the paper.
>
> **Weakness 5.**
> We will add some discussion on applying our algorithm to high-dimensional data in the final version of the paper.

---

> > ### Comment · Reviewer_eUVC · 2023-08-17
> >
> > Thank you for your answers. The additional experiment (from Table 1) looks good, a larger eigengap than SZ seems to imply a better sparsification result compared to SZ. I have no further questions.

---

### Official Review · Reviewer_awKP · 2023-07-07

**Soundness:** 4 excellent
**Presentation:** 4 excellent
**Contribution:** 4 excellent
**Rating:** 8
**Confidence:** 3

**Summary:**

Authors proposes a fast approximation method for constructing sparse similarity graphs from data for spectral clustering. They leverage Sun and Zanetti (2019)'s sampling-based approach to find "cluster-preserving sparsifier" for similarity graphs. This is combined with black box KDE and binary search algorithms to generate a fast & memory efficient approximation of SZ's algorithm. The experimental results are shown to be fast and accurate compared to Scikit-learn and FAISS's implementation.

**Strengths:**

This paper combines existing ideas (e.g. SZ algorithm, KDE algorithm, binary search) to create a novel approach to kernel graph construction that has potential to be used by many researchers in the community. The experimental results including runtime comparison and image segmentation (e.g. Figure 2) are quite impressive. It was also well written and clearly presented.

**Weaknesses:**

Please see questions below.

**Questions:**

1. Given how it was used as a major motivation in this paper, I expected to see a comparison with the output of SZ algorithm in the experiments section. Have the authors compared SZ and proposed method experimentally for clustering?

2. What do the calculated probabilities and sampled sparse graphs look like? Can the authors show any examples on a small graph or using a synthetic dataset? And how does the graph actually compare to that from SZ?

3. It's missing a discussion on limitations of this work.

**Limitations:**

Not really.

---

> ### Author Rebuttal · Authors · 2023-08-09
>
> Many thanks for your very positive report.   Here are our responses to your questions.
>
> **Question 1.**
> In Figure 1 of the attached PDF file, we provide an updated version of the experiment on the two moons dataset to include the results of the SZ algorithm.
> The SZ algorithm iterates over every edge weight in the fully connected similarity graph in order to construct a sparsifier, and so its running time has a quadratic dependency on the number of data points.
>
> **Question 2.**
> Figure 2 of the attached PDF file illustrates the structure of the constructed graph on a small toy dataset with 100 data points.
> We observe that both graphs preserve the cluster structure of a fully connected similarity graph.
> We will include an illustration demonstrating the sparse graph structure in the final version of the paper.
>
> **Question 3.** We will add more discussion on the limitations of our work in the final version of the paper.

---

> > ### Comment · Reviewer_awKP · 2023-08-17
> >
> > Thank you for the comments and additional experimental results. I acknowledge that I've read comments from the authors and other reviewers, and have no further questions.

---

### Author Rebuttal · Authors · 2023-08-09

We thank all reviewers for their positive and detailed reviews. We respond to their specific questions individually, with reference to some additional figures and table in the attached PDF.

---

### Decision · Program_Chairs · 2023-09-21

**Decision:**

Accept (spotlight)

**Comment:**

This paper proposes an efficient approach for similarity graph construction based on kernel density estimation. The reviewers appreciate the theoretical soundness and high originality of the proposed approach. Since all the reviewers are positive for the paper, I recommend accepting the paper.